# LIPSCHITZ-GUIDED MONTE CARLO TREE SEARCH WITH KNOWLEDGE TRANSFER ACROSS SEQUENTIAL TASKS

## ABSTRACT

Monte Carlo Tree Search (MCTS) has proven highly effective in solving complex planning tasks by balancing exploration and exploitation using Upper Confidence Bound for Trees (UCT). However, existing works have not considered MCTS-based lifelong planning facing a sequence of MDPs – e.g., each MDP with varying transition probabilities and rewards from previous ones – throughout the operational lifetime. This paper presents LiZero for Lipschitz lifelong planning using MCTS. We propose a novel concept of adaptive UCT (aUCT) to transfer knowledge from previous tasks to the exploration/exploitation of a new task, depending on both the Lipschitz continuity between tasks and the confidence of knowledge in Monte Carlo action sampling. We analyze LiZero's acceleration factor in terms of improved sampling efficiency and also develop efficient algorithms to compute aUCT in an online fashion by both data-driven and model-based approaches, whose sampling complexity and error bounds are also characterized. Numerical results show that LiZero significantly outperforms existing MCTS and lifelong learning baselines in terms of much faster convergence ($3\sim4$x). Our results highlight the potential of LiZero to advance decision-making and planning in dynamic environments.

## 1 INTRODUCTION

Monte Carlo Tree Search (MCTS) has demonstrated state-of-the-art performance in solving many challenging planning tasks, from playing the game of Go Silver et al. (2016) and chess to logistic planning Silver et al. (2017). It performs look-ahead searches based on Monte Carlo sampling of the actions to balance efficient exploration and optimized exploitation in the large search space. Recent efforts have focused on developing MCTS algorithms for real-world domains that require the elimination of certain standard assumptions. Examples include MuZero Schrittwieser et al. (2020b) that leverages the decoding of hidden states to avoid requiring the knowledge of the game dynamics; and MAzero Liu et al. (2024) that performs multi-agent search through decentralized execution. Existing work have not considered MCTS with lifelong task-to-task variations, by retaining and transferring prior knowledge to bootstrap lifelong learning across a continuous stream of tasks.

We consider MCTS-based lifelong planning under task-to-task variation. An agent faces a series of changing planning tasks – e.g., with varying transition probabilities and rewards – which are drawn sequentially throughout the operational lifetime. Transferring knowledge from prior experience to continually adapt Monte Carlo sampling of the actions and thus speed up searches in new tasks is a key question in this setting. We note that although continual and lifelong planning has been studied in reinforcement learning (RL) context, e.g., learning models of the a task sequence of distinct stationary environments Xie et al. (2020), identifying reusable skills Lu et al. (2020), or estimating Bayesian sampling posteriors Fu et al. (2022), such prior works do not apply to MCTS. Monte Carlo action sampling in MCTS relies on Upper Confidence Tree (UCT) or Predictor Upper Confidence Tree (pUCT) Auger et al. (2013); Matsuzaki (2018) to balance exploration and exploitation in large search spaces. To the best of our knowledge, there has not been existing work analyzing the transfer of knowledge from past MCTS searches to new tasks, thus enabling adaptive UCT/pUCT rules in lifelong MCTS.

This paper proposes LiZero for Lipschitz lifelong planning using MCTS. We quantify a novel concept that the amount of knowledge transferable from a source task to the UCT/pUCT rule of a new task depends on both the similarity between the tasks as well as the confidence of the knowledge. More precisely, by defining a distance metric between two MDPs, we refine the concentration argument and drive a new adaptive UCT bound (denoted as aUCT in this paper) for lifelong MCTS. The aUCT is shown to consist of two components – relating to (i) the Lipschitz continuity between the two tasks and (ii) the confidence of knowledge due to the number of samples in Monte Carlo action sampling. Our results enable the development of a novel LiZero algorithm that makes use of prior experience to run an adaptive MCTS by simulating/traversing from the root node and selecting actions according to the aUCT rule, until reaching a leaf node. We also analyze aUCT's acceleration factor in terms of improved sampling efficiency due to cross-task transfer. It is shown that smaller task distance and higher confidence can both lead to higher acceleration in aUCT.

To support the practical deployment of LiZero in lifelong planning, we need efficient solutions to compute aUCT in an online fashion. To this end, we develop practical algorithms to estimate various terms in aUCT, and especially the distance metric between two MDPs, from either available state-action samples using a data-driven approach or a parameterized distance using a model-based (deep learning) approach. We provide rigorous analysis on the sampling complexity of the data-driven approach, to ensure arbitrarily small errors with high probability, by modeling a sequence of policy update process by a filtration – i.e., an increasing sequence of $\sigma$-algebras. For the model-based approach, we obtain an upper bound using a parameterized distance of the neural network models. These results enable effective LiZero applications to open world tasks. We evaluate LiZero on a series of learning tasks with varying transition probabilities and rewards. LiZero is shown to significantly outperform MCTS and lifelong RL baselines (e.g., Winands (2024); Kocsis & Szepesvári (2006); Cheng et al.; Schrittwieser et al. (2020a); Brafman & Tennenholtz (2002); Lecarpentier et al. (2021a)) in terms of performance. faster convergence to higher optimal rewards. Using the knowledge of only a few source tasks, LiZero achieves 3∼4x speedup with about 31% higher early reward in the first half of the learning process.

Our key contributions are as follows. First, we study theoretically the transfer of past experience in MCTS and develop a novel aUCT rule, depending on both Lipschitz continuity between tasks and the confidence of knowledge in Monte Carlo action sampling. It is proven to provide positive acceleration in MCTS due to cross-task transfer. Second, we develop LiZero for lifelong MCTS planning, with efficient methods for online estimation of aUCT and analytical error bounds. Finally, LiZero achieves significant speed-up over MCTS and lifelong RL baselines in lifelong planning.

## 2 BACKGROUND

Monte Carlo Tree Search (MCTS) Kocsis & Szepesvári (2006); Silver et al. (2016); Schrittwieser et al. (2020b) is a heuristic algorithm for solving problems modeled as Markov Decision Processes (MDPs). It dynamically balances exploration and exploitation by expanding a search tree, simulating action outcomes, and updating value estimates accordingly. An MDP is typically defined as $\langle \mathcal{S}, \mathcal{A}, R, P \rangle$, where $\mathcal{S}$ and $\mathcal{A}$ are the state and action spaces, $R_s^a$ is the reward, and $P$ denotes the transition dynamics.

In the MCTS framework, Upper Confidence Bound for Trees (UCT) Coulom (2006) and its variant, Predictor UCT (pUCT) Matsuzaki (2018); Auger et al. (2013), are widely used to balance exploration and exploitation during node selection. While effective under static assumptions, they perform suboptimally in dynamic, a task sequence of distinct stationary environments where state transitions and rewards change over time Pourshamsaei & Nobakhti (2024); Hernandez-Leal et al. (2017); Goldberg & Matarić (2003). In this paper, we consider MCTS- based lifelong planning, where an agent faces a sequence of distinct MDPs – e.g., with varying transition probabilities and reward – and requires the development of new adaptive UCT bounds.

Lifelong reinforcement learning (RL) Lecarpentier et al. (2021b); Xie et al. (2020); Fu et al. (2022); Lu et al. (2020); Auger et al. (2013) addresses learning a sequence of tasks from unknown MDPs in an online manner. Each sampled MDP is treated as a standalone RL problem where the agent learns and adapts its policy $\pi$ to maximize expected return Da Silva et al. (2018); Hawasly & Ramamoorthy (2013); Abel et al. (2018). We can reasonably believe that the knowledge gained in similar MDPs can be reused. While prior work explores task modeling Xie et al. (2020), skill reuse Lu et al. (2020), and

Bayesian transfer Fu et al. (2022), these methods do not extend naturally to lifelong MCTS, which requires adaptive UCT-style bounds to enable knowledge transfer and efficient planning across a sequence of distinct tasks (MDPs).

## 3 OUR PROPOSED SOLUTION

### 3.1 DERIVING ADAPTIVE UPPER CONFIDENCE BOUND (aUCT)

To derive the proposed aUCT rule, we consider a set of $m$ past known MDPs $\mathcal{M}_1, \ldots, \mathcal{M}_m$ and their learned search policies $\pi_1, \ldots \pi_m$. Let $S$ and $A$ be their state and action spaces, respectively[1], $N_i(s, a)$ be the visit count of MDP $\mathcal{M}_i$ to state-action pair $(s \in S, a \in A)$, $W_i(s, a)$ to denote its sampled return, and $Q_{\mathcal{M}_i}^{N_i}(s, a) = W_i(s, a)/N_i(s, a)$ be the learned estimate for the Q-value of MDP $\mathcal{M}_i$. Our goal is to apply these knowledge toward learning a new MDP, denoted by $\mathcal{M}$. To this end, we derive a new Lipschitz upper confidence bound for $\mathcal{M}$, which utilizes and transfers the knowledge from past MDPs $\mathcal{M}_1, \ldots, \mathcal{M}_N$, thus obtaining an improved Monte Carlo action sampling strategy that limits the tree search in $\mathcal{M}$ to a smaller subset of sampled actions. We use $N(s, a)$ to denote the visit count of the new MDP to $(s \in S, a \in A)$, $W(s, a)$ to denote the sampled return, and thus $Q_{\mathcal{M}}^N(s, a) = W(s, a)/N(s, a)$ to denote its current Q-value estimate.

Our key idea in this paper is that an improved upper confidence bound for the new MDP $\mathcal{M}$ can be obtained by (i) analyzing the Lipschitz continuity between the past and new MDPs with respect to the upper confidence bounds and (ii) taking into account the confidence and aleatory uncertainty of the learned Q-value estimates to determine to what extent the learned knowledge from each $\mathcal{M}_i$ is pertinent. Intuitively, the more similar $\mathcal{M}$ and $\mathcal{M}_i$ are and the more samples (and thus higher confidence) we have in the learned Q-value estimates, the less exploration we would need to perform to solve $\mathcal{M}$ through MCTS. Our analysis will lead to an improved upper confidence bound that guides the MCTS on the new MDP $\mathcal{M}$ over a much smaller subset of action samples, thus significantly improving search performance. We start by introducing a definition of the distance between two given MDPs, $\mathcal{M} = \langle R, P \rangle$, $\mathcal{M}' = \langle R', P' \rangle$, with reward functions $R, R'$ and state transitions $P, P'$, respectively. We choose a positive scaling factor $\kappa > 0$ to combine the distances for transition probabilities and rewards. Proofs of all theorems and corollaries are presented in the appendix.

**Definition 3.1.** Given two MDPs $\mathcal{M} = \langle R, P \rangle$, $\mathcal{M}' = \langle R', P' \rangle$, and a distribution for sampling the state transitions $\mathcal{U} : \mathcal{S} \times \mathcal{A} \times \mathcal{S}' \to [0, 1]$, we define the pseudometric between the MDPs as:

$$d(\mathcal{M}, \mathcal{M}') = \Delta R + \kappa \cdot \Delta P = \mathbb{E}_{(s,a,s') \sim \mathcal{U}} \left[ |R_s^a - R_s'^a| + \kappa |P_{ss'}^a - P_{ss'}'^a| \right].$$

Noted that we write $d(\mathcal{M}, \mathcal{M}') = \Delta R + \kappa \cdot \Delta P$ with $\Delta_R \in [0, R_{\max}]$ and $\Delta_P \in [0, 1]$(total-variation distance), so $d \in [0, R_{\max}] + \kappa$. The normalized advantage $\Delta_{(s,a)}^M := \frac{Q_M^*(s, a^*) - Q_M^*(s, a)}{R_{\max}/(1-\gamma)} \in [0, 1]$ $\kappa$ is a constant used to remove the mismatch in units between $\Delta R$ and $\Delta P$, the detailed derivation is given in Eq. 18.

Here $d(\mathcal{M}, \mathcal{M}')$ is our definition of distance between two MDPs, $\mathcal{M}$ and $\mathcal{M}'$. We choose $\mathcal{U}$ to be a uniform distribution for sampling the state transitions in this paper. In Section 4, we discuss practical algorithms to estimate the distance metric between two MDPs, from either available state-action samples using a data-driven approach or a parameterized distance using a model-based (deep learning) approach. The sampling complexity and error bounds are also analyzed.

Next, we prove the main result of this paper and show that the upper confidence bounds of $\mathcal{M}$ and $\mathcal{M}'$ are Lipschitz continuous with respect to distance $d(\mathcal{M}, \mathcal{M}')$. We obtain a new upper confidence bound for $\mathcal{M}$, by transfering the knowledge from the learned Q-value estimates $Q_{\mathcal{M}'}^{N'}(s, a) = W'(s, a)/N'(s, a)$ of MDP $\mathcal{M}'$. The bound also depends on the confidence of learned Q-value estimates, relating to the visit counts $N(s, a)$ and $N'(s, a)$.

**Theorem 3.2** (Lipschitz aUCT Rule). *Consider two MDPs $M$ and $M'$ with visit count $N, N'$ and corresponding estimate Q-values $Q_M^N(s, a), Q_{M'}^{N'}(s, a)$, respectively. With probability at least $(1 - \delta)$ for some positive $\delta > 0$, we have*

$$\left| Q_{\mathcal{M}}^N(s, a) - Q_{\mathcal{M}'}^{N'}(s, a) \right| \leq L \cdot d(\mathcal{M}, \mathcal{M}') + P(N, N') \tag{1}$$

---

[1]Without loss of generality, we assume that the MDPs have the same state and action spaces. Otherwise, we can consider the extended MDPs defined by the union of their state and action spaces.

where $L = 1/(1 - \gamma)$ *(From Eqn 18) is a Lipschitz constant, $d(\mathcal{M}, \mathcal{M}')$ is the distance between MDPs, and $P(N, N')$ is given by*

$$P(N, N') = \frac{2R_{\max}}{1 - \gamma} \sqrt{\frac{\ln(2/\delta)}{2 \cdot \min(N, N')}} \tag{2}$$

In the theorem above, we show that the estimate Q-values between two MDPs are bounded by two terms, i.e., a Lipschitz continuity term depending on the distance $d(\mathcal{M}, \mathcal{M}')$ between the two environments and a confidence term depending on the number $N, N'$ of samples used to estimate the Q-values. The Lipschitz continuity term measures how much the learned knowledge of source MDP $\mathcal{M}$ is pertinent to the new MDP $\mathcal{M}'$, while the confidence terms $P(N, N')$ quantifies the sampling bias arising from statistical uncertainty due to limited sampling in MCTS. We note that as the number of samples $N$ goes to infinity, we have $Q_{\mathcal{M}}^N(s, a) \rightarrow Q_{\mathcal{M}}^*(s, a)$ in Theorem 3.2, approaching the true Q-value $Q_{\mathcal{M}}^*(s, a)$ of the new MDP. Our theorem effectively provides an upper confidence bound for the true Q-value of the new MDP, based on knowledge transfer from the source MDP. We also note that as both numbers $N, N'$ go to infinity, the confidence term becomes $P(N, N') \rightarrow 0$. Our theorem recovers the Lipschitz lifelong RL Lecarpentier et al. (2021b) as a special case of our results, for the true Q-values of the two MDPs.

We apply Theorem 3.2 to MCTS-based lifelong planning with a sequence of distinct $m$ tasks, $\mathcal{M}_1, \ldots, \mathcal{M}_m$. Our goal is to obtain an improved bound on the true Q-value of the new task $\mathcal{M}$ based on knowledge transfer. To this end, we independently apply the knowledge from each past MDP, i.e., $Q_{\mathcal{M}_i}^{N_i}(s, a) = W_i(s, a)/N_i(s, a)$, to the new MDP. By taking the minimum of these bounds and making $N \rightarrow \infty$, it provides a tightest upper bound on the true Q-value $Q_{\mathcal{M}}^*(s, a)$ of the new MDP, which is defined as our aUCT bound, as it adaptively transfers knowledge from past tasks to the new tasks in MCTS-based lifelong planning. The result is summarized in the following corollary.

**Corollary 3.3** (aUCT bound in lifelong planning). *Given MDPs $\mathcal{M}_1, \ldots, \mathcal{M}_m$, the new MDP's true Q-value is bounded by $Q_{\mathcal{M}}^*(s, a) \leq U_{\mathrm{aUCT}}$ with probability at least $(1 - \delta)$. The aUCT bound $U_{\mathrm{aUCT}}$ is given by*

$$U_{\mathrm{aUCT}}(s, a) \triangleq \min_{1 \leq i \leq m} \left[ Q_{M_i}^{N_i}(s, a) + L \cdot d(\mathcal{M}, \mathcal{M}_i) + \frac{2R_{\max}}{1 - \gamma} \sqrt{\frac{\ln(2/\delta)}{2N_i(s, a)}} \right] \tag{3}$$

Obtaining this corollary is straightforward from Theorem 3.2 by taking $N \rightarrow \infty$ and considering the tightest bound of all knowledge transfers. In the context of MCTS-based lifelong planning, the more knowledge we have from solving past tasks, the more likely we can easily plan a new task, as the aUCT bound $U_{\mathrm{aUCT}}(s, a)$ is taken over the minimum of all past tasks. The confidence of past knowledge, i.e., the statistical uncertainty due to sampling number $N_i$, also affects the knowledge transfer to the new task.

## 3.2 OUR PROPOSED LIZERO ALGORITHM USING aUCT

We use the derived aUCT to design a highly efficient LiZero algorithm for MCTS-based lifelong planning. The LiZero algorithm transfers knowledge from past known tasks by computing $U_{\mathrm{aUCT}}(s, a)$ in Corollary 3.3. It requires an efficient estimate of the distance $d(\mathcal{M}, \mathcal{M}_i)$ (as defined in Definition 3.1) between the source MDPs and the new (target) MDP. We will present practical algorithms for such distance estimates in the next section and present an analysis of the sampling complexity and error bounds. We will first introduce our LiZero algorithm in this section. We note that, during MCTS, direct exploration/search in the new task $\mathcal{M}$ also produces new knowledge and leads to improved UCT bound of $\mathcal{M}$. Therefore, our proposed LiZero combines both knowledge transfer through $U_{\mathrm{aUCT}}(s, a)$ and knowledge from direct exploration/search in $\mathcal{M}$.

The search in our proposed LiZero algorithm is divided into three stages, repeated for a certain number of simulations. First, each simulation starts from the internal root state and finishes when the simulation reaches a leaf node. Let $Q_{\mathcal{M}}^N(s, a) = W(s, a)/N(s, a)$ be the current estimate of the new MDP and $N(s) = \sum_{a \in \mathcal{A}} N(s, a)$ be the visit count to state $s \in \mathcal{S}$. For each simulated time step, LiZero chooses an action $a$ by maximizing a combined upper confidence bound based on aUCT, i.e.,Since both terms are valid upper bounds on $Q^*(s, a)$, their minimum remains an admissible upper

bound, preserving optimism. We ensure sufficient exploration by standard UCB tie-breaking and a per-edge visit floor.

$$a = \arg\max_a \min \left[ \frac{W(s,a)}{N(s,a)} + C\sqrt{\frac{\ln N(s)}{N(s,a)}}, U_{\text{aUCT}}(s,a) \right]$$

In practice, we can also use the maximum possible return $R_{\max}/(1-\gamma)$ as an initial value of the search. Next, at the final time step of the simulation, the reward and state are computed by a dynamics function. A new node, corresponding to the leaf state, is then added to the search tree. Finally, at the end of the simulation, the statistics along the trajectory are updated. Let $G$ be the accumulative (discounted) reward for state-action $(s,a)$ from the simulation. We update the statistics by: $Q_{\mathcal{M}}^{N+1}(s,a) := \frac{N(s,a) \cdot Q_{\mathcal{M}}^{N}(s,a) + G}{N(s,a)+1}$, $\quad N(s,a) := N(s,a) + 1$.

**No negative transfer.** By construction, LiZero always selects $a^*$ Therefore, if the transferred bound $U_{\text{aUCT}}(s,a)$ is loose (e.g., when tasks are dissimilar or Lipschitz regularity does not hold), the minimum simply reduces to the standard UCT confidence term, and LiZero behaves identically to vanilla UCT. In particular, Theorem 3.4 shows that the sample complexity of LiZero is never worse than that of UCT ($\Gamma \geq 1$), and equals it ($\Gamma = 1$) when no useful transfer is available.

Intuitively, at the start of task $\mathcal{M}$'s MCTS, there are not sufficient samples available, and thus $U_{\text{aUCT}}(s,a)$ serves as a tighter upper confidence bound than that resulted from the Monte Carlo actions sampling in $\mathcal{M}$. As more samples are obtained during the search process, the standard UCT bound is expected to become tighter than $U_{\text{aUCT}}(s,a)$. Using both bounds will ensure efficient knowledge transfer and task-specific search. The pseudocode of LiZero is provided in Appendix A.2.

For the proposed LiZero algorithm, we prove that it can result in accelerated convergence in MCTS. More precisely, we analyze the sampling complexity for the learned Q-value estimate $Q_{\mathcal{M}}^{N}(s,a)$ to converge to the true value $Q_{\mathcal{M}}^{*}(s,a)$, and demonstrate a strictly positive acceleration factor, compared to the standard UCT. The results are summarized in the following theorem.

**Theorem 3.4.** *To ensure the convergence in a finite state-action space,* $\max_{(s,a)} |Q_{\mathcal{M}}^{N}(s,a) - Q_{\mathcal{M}}^{*}(s,a)| \leq \epsilon$ *with probability* $1 - \delta$, *the number of samples required by standard UCT is*

$$\tilde{O}\left( \frac{|\mathcal{S}| \cdot |\mathcal{A}|}{(1-\gamma)^3 \epsilon^2} \ln \frac{1}{\delta} \right), \tag{4}$$

*while the proposed LiZero algorithm requires:* $\tilde{O}\left( \frac{1}{\Gamma} \cdot \frac{|\mathcal{S}| \cdot |\mathcal{A}|}{(1-\gamma)^3 \epsilon^2} \ln \frac{1}{\delta} \right)$. *where* $\Gamma > 1$ *is an acceleration factor given by* $\Gamma = \frac{\sum_{(s,a) \in \mathcal{S}_1 \cup \mathcal{S}_0} \frac{1}{(\Delta_{(s,a)}^{\mathcal{M}})^2}}{\sum_{(s,a) \in \mathcal{S}_1} (1) + \sum_{(s,a) \in \mathcal{S}_0} \frac{1}{(\Delta_{(s,a)}^{\mathcal{M}})^2}}$, *and* $\mathcal{S}_1 = \{(s,a) \mid \exists i : U_{\text{aUCT}}(s,a) < Q_{\mathcal{M}}^{*}(s,a^*)\}$ *is a state-action set where* $U_{\text{aUCT}}$ *of action* $a$ *is lower than the optimal return of* $a^*$ *in state* $s$; *and* $\Delta_{(s,a)}^{\mathcal{M}} \propto [Q_{\mathcal{M}}^{*}(s,a^*) - Q_{\mathcal{M}}^{*}(s,a)]$ *is a normalized advantage in the range of* $[0,1]$.

The theorem shows that LiZero achieves a strictly improved acceleration $\Gamma > 1$ with a reduced sampling complexity (by $1/\Gamma$), in terms of ensuring convergence to the optimal estimates, i.e., $\max_{(s,a)} |Q_{\mathcal{M}}^{N}(s,a) - Q_{\mathcal{M}}^{*}(s,a)| \leq \epsilon$ with probability $1 - \delta$. Since the normalized advantage $\Delta_{(s,a)}^{\mathcal{M}}$ is in $[0,1]$, we have $1/\Delta_{(s,a)}^{\mathcal{M}} \geq 1$. It follows that $\Gamma > 1$ whenever $\mathcal{S}_1 \neq \emptyset$; when $\mathcal{S}_1 = \emptyset$, LiZero reduces to UCT and $\Gamma = 1$. More precisely, LiZero achieves higher acceleration when (i) our $aUCT$ makes more actions $a$ less favorable, as $U_{\text{aUCT}}(s,a) < Q_{\mathcal{M}}^{*}(s,a^*)$ implies that the sub-optimality of action $a$ in $s$ can be more easily determined due to aUCT; or (ii) $aUCT$ helps establish tighter bounds in cases with a smaller advantage, which naturally requires more samples to distinguish the optimal actions – since $\Gamma$ increases as the normalized advantage becomes smaller for $(s,a) \in \mathcal{S}_1$, while being larger for $(s,a) \in \mathcal{S}_0$. These explain LiZero's ability to achieve much higher acceleration and lower sampling complexity, resulted from significantly reduced search spaces. We will evaluate this acceleration/speedup through experiments in Section 5.

## 4 ESTIMATING aUCT IN PRACTICE

To deploy LiZero in practice, we propose two approaches for estimating aUCT, and in particular, the distance $d_{\mathcal{M}, \mathcal{M}_i}$ between two MDPS. Our first approach leverages trajectory samples drawn

from MCTS policies by modeling a sequence of distict policies as a filtration – i.e., an increasing sequence of $\sigma$-algebra, while our second approach learns neural network approximations of the MDPs. Analysis of sampling complexity and error bounds are provided as theorems in this section.

**Sample-based Distance Estimate.** During MCTS, transition samples are collected from the search to train a search policy $\pi$. It is easy to see that we can leverage these transition samples to estimate distance $d(\mathcal{M}, \mathcal{M}')$ between two MDPs, as long as we address the bias arising from gap between search policy $\pi$ and desired sampling distribution $\mathcal{U}$ in the distance definition $d(\mathcal{M}, \mathcal{M}')$. It also allows us to obtain a consistent estimate of MDP distance, without depending on the search policy that is updated during training. We note that this bias can be addressed by importance sampling.

Let $\Delta X(s,a) = \Delta R_s^a + \kappa \Delta P_s^a$ be the distance metric for a given state-action pair $(s,a)$. We can rewrite the distance as $d(\mathcal{M}, \mathcal{M}') = \mathbb{E}_{(s,a)\sim\mathcal{U}}[\Delta X(s,a)]$. We denote $p_{\mathcal{U}}(s,a)$ as the probability (or density) of sampling $(s,a)$ according to distribution $\mathcal{U}$. Importance sampling implies:

$$\mathbb{E}_{(s,a)\sim\mathcal{U}}[\Delta X(s,a)] = \mathbb{E}_{(s,a)\sim\pi}\left[\frac{p_{\mathcal{U}}(s,a)}{\pi(s,a)} \cdot \Delta X(s,a)\right], \tag{5}$$

which can be readily computed from the collected transition samples, following the search policy $\pi(s,a)$. Therefore, for a given set of samples $\{(s_i, a_i), \forall i = 1, \ldots, n\}$ collected from a search policy $\pi(s,a)$, we can estimate the distance by the empirical mean:

$$\hat{d}_1 = \frac{1}{n}\sum_{i=1}^{n} w_i \Delta X(s_i, a_i), \text{ with } w_i = \frac{\mathcal{U}(s_i, a_i)}{\pi(s_i, a_i)} \tag{6}$$

where $w_i$ is the importance sampling weight.

As long as the state-action pairs with $\pi(s,a) > 0$ cover the support of $\mathcal{U}$, this estimator satisfies $\mathbb{E}[\hat{d}_\infty] = d(\mathcal{M}, \mathcal{M}')$, meaning it is unbiased. Let $\alpha$ be the "coverage" of policy $\pi(s,a)$, i.e. $\pi(s,a) \geq \alpha > 0$, and let $p_{\mathcal{U}}^{\max}$ be the maximum desired sampling probability. We summarize this result in the following theorem and state the sampling complexity for estimator $\hat{d}_1$ to $\epsilon$-converge to $d(\mathcal{M}, \mathcal{M}')$.

In our tabular gridworld experiments, both the state and action spaces are finite. We instantiate the reference distribution $U$ in Definition 3.1 as the uniform distribution over state–action pairs, $U(s,a) = \frac{1}{|\mathcal{S}||\mathcal{A}|}$. The importance weights in LiZero-P are then given by $w_i = \frac{U(s_i,a_i)}{\pi(s_i,a_i)}$, where $\pi$ denotes the empirical tree policy estimated from visit counts under UCT-style exploration. In the finite gridworlds, every action at a visited state has a strictly positive lower bound on $\pi(s,a)$, so the weights $\{w_i\}$ are almost surely bounded and we observe no variance explosion in practice. For larger or continuous domains, standard techniques such as self-normalized importance sampling and weight clipping can be incorporated without changing the theoretical analysis.

**Theorem 4.1** (Sampling Complexity under Stationarity). *Assume that for any $(s,a)$, the reward plus transition difference is bounded, i.e., $\Delta X(s,a) \in [0,b]$, and that there exists $\alpha$ such that $\pi(s,a) \geq \alpha > 0$. When $n$ independent samples are used to estimate $\hat{d}_1$, we have*

$$Pr\{|\hat{d}_1 - d(\mathcal{M}, \mathcal{M}')| \leq \epsilon\} \geq 1 - \delta \tag{7}$$

for any $\delta \in (0,1)$, if the number of samples satisfy $n \geq \frac{1}{2\epsilon^2}b^2\left(\frac{p_{\mathcal{U}}^{\max}}{\alpha}\right)^2 \cdot \ln\left(\frac{2}{\delta}\right)$ Thus, we obtain a convergence guarantee in the sense of arbitrarily high probability $1 - \delta$ and arbitrarily small error $\epsilon$, for estimating $d(\mathcal{M}, \mathcal{M}')$ using $\hat{d}_1$. $\hat{d}_1$ is unbiased and ensures convergence to the true distance as the number of samples is sufficiently large.

We note that in many practical settings, the search policy $\pi$ would not stick to a stationary distribution. In contrast, it is continuously updated in each iteration, resulting in a sequence of distinct policies over time, i.e., $\pi_1, \pi_2, \ldots, \pi_k$. Thus, the transition samples $(s_k, a_k)$'s we obtain at each step $k$ for estimating the distance $d(\mathcal{M}, \mathcal{M}')$ are indeed drawn from a different $\pi_k$. We cannot assume that the samples follow a stationary distribution (nor that $\{\Delta X_k^w\}$ are i.i.d.) in importance sampling. we model the sequence of distinct policy updates as a filtration – i.e., an increasing sequence of $\sigma$-algebra. In particular, we make the following assumption: at the $k$-th sampling step, the environment is

forcibly reset to a predetermined policy $\pi_k$ or independently draws a state from an external memory. This assumption is reasonable, in many episodic learning scenarios, the environment is inherently divided into episodes: at the beginning of each episode, the state is reset to some initial distribution (e.g., the opening state in Atari games). This naturally results in the "reset" assumption.

In this setup, the policy $\pi_k$ at step $k$ is determined by information at step $k-1$ or earlier. Consequently, once $\pi_k$ is fixed, the distribution (marginal) of $\Delta X_k^w = \frac{p_\mathcal{U}(s_k,a_k)}{\pi_k(s_k,a_k)} \Delta X(s_k,a_k)$ is also fixed. Therefore, we can establish the filtration $\{\mathcal{F}_k, k = 1, 2, \ldots\}$ as follows:

$$\mathcal{F}_{k-1} = \sigma\{\pi_1, ..., \pi_k, (s_1, a_1), ..., (s_{k-1}, a_{k-1})\}, \tag{8}$$

where $\sigma\{\cdot\}$ denotes the smallest $\sigma$-algebra generated by the random elements. Thus, we obtain:

$$\mathbb{E}[\Delta X_k | \mathcal{F}_{k-1}] = \mathbb{E}_{(s_k,a_k) \sim \pi_k} \left[ \frac{p_\mathcal{U}(s_k,a_k)}{\pi_k(s_k,a_k)} \cdot \Delta X(s_k, a_k) \right] = \mathbb{E}_{(s_k,a_k) \sim \mathcal{U}}[\Delta X(s,a)] = d(\mathcal{M}, \mathcal{M}') \tag{9}$$

This allows us to obtain another empirical estimator $\hat{d}_2$ using the filtration model. We analyze the sampling complexity of $\hat{d}_2$ and summarise the results in the following theorem.

**Theorem 4.2** (Sampling Complexity under task-to-task variation). *Under the same conditions as Theorem 4.1 when $n$ independent samples are used to estimate $\hat{d}_2$, we have $Pr\{|\hat{d}_2 - d(\mathcal{M}, \mathcal{M}')| \leq \epsilon\} \geq 1 - \delta$ for any $\delta \in (0, 1)$, if the number of samples satisfy $n \geq \frac{2}{\epsilon^2} b^2 \left( \frac{p_\mathcal{U}^{\max}}{\alpha} \right)^2 \cdot \ln\left( \frac{2}{\delta} \right)$.*

It implies that more samples are needed, considering the task-to-task variation of the policy update process for the distance estimate. We approximate the filtration by resetting episodes and re-sampling initial states from a replay buffer at task boundaries.

**NN-based Distance Estimate.** We propose an alternative approach to first approximate the dynamics of MDPs $\mathcal{M}$ and $\mathcal{M}'$ using two neural networks and then estimate $d(\mathcal{M}, \mathcal{M}')$ based on the parameterized distance between the neural networks. To this end, we need to establish a bound on $d(\mathcal{M}, \mathcal{M}')$ using the distance between their neural network parameters. We use a neural network $\Psi_\phi : \mathcal{S} \times \mathcal{A} \to \Delta(\mathcal{S})$ to model the MDP dynamics. Many model-based learning algorithms, such as PILCO Deisenroth & Rasmussen (2011),MBPO Janner et al. (2019),PETS Chua et al. (2018),MuZero Schrittwieser et al. (2020b), can be employed to learn the models of $\mathcal{M}$ and $\mathcal{M}'$. Let $\phi$ be the neural network parameters of MDP $\mathcal{M}$ and $\phi'$ be the neural network parameters of MDP $\mathcal{M}'$. We define a distance in the parameter space: $\hat{d}_{para} = \rho(\phi, \phi') \geq 0$, where $\rho$ is a distance or divergence measure in the parameter space, such as the $\ell_2$-norm or certain kernel distances. Intuitively, if $\phi$ and $\phi'$ are very close, the two neural networks are similar in fitting the dynamics of the respective MDPs. It suggests that the two MDPs should have a small distance. To provide a more rigorous characterization of this concept, we present the following theorem, which demonstrates that under proper assumptions, the distance $\hat{d}_{para}$ based on neural network parameters can serve as an upper bound for the desired $d(\mathcal{M}, \mathcal{M}')$. Let $\kappa = R_{\max}\gamma/(1 - \gamma)$ be a constant.

**Theorem 4.3.** *If the neural networks modeling $\mathcal{M}$ and $\mathcal{M}'$ satisfy the Lipschitz condition, i.e., there exists a constant $L > 0$ such that $\forall(s, a)$, $\|\Psi_\phi(s, a) - \Psi_{\phi'}(s, a)\|_1 \leq L \cdot \rho(\phi, \phi')$, then we have: $d(\mathcal{M}, \mathcal{M}') \leq (1 + \kappa)L\hat{d}_{para}$.*

The theorem indicates that by learning neural networks to model the MDP dynamics, we can estimate the distance $d(\mathcal{M}, \mathcal{M}')$ by estimating the distance between the neural network parameters. This parameterized distance can be computed for event continuous action and state spaces.

## 5 EVALUATIONS

Our experiments evaluate LiZero on a series of ten learning tasks with varying transition probabilities and rewards. We demonstrate LiZero's ability to transfer past knowledge in MCTS-based planning, resulting in significant convergence speedup (3~4x) and early reward improvement (about 31% average improvement during the first half of learning process) in lifelong planning problems. All experiments are conducted on a Linux machine with AMD EPYC 7513 32-Core Processor CPU and an NVIDIA RTX A6000 GPU, implemented in python3. All source codes are made available in the supplementary material.

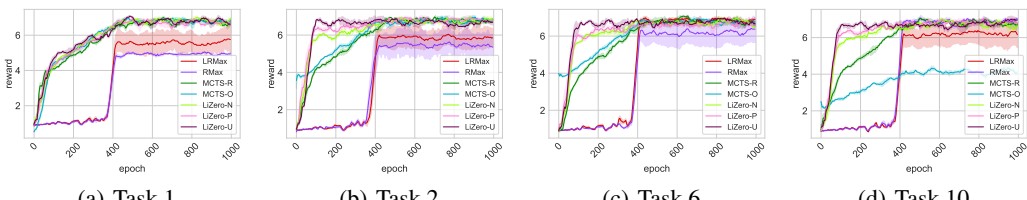

| (a) Task 1 | (b) Task 2 | (c) Task 6 | (d) Task 10 |

Figure 1: Comparing LiZero with MCTS and lifelong RL baselines. We demonstrate the convergence of different algorithms on representative Tasks 1, 2, 6, and 10, in a sequence of distinct ten tasks. In Task 1, since no prior knowledge is yet available, our LiZero and other MCTS baselines show similar convergence speeds and optimal rewards. From Task 2 to Task 10, as more knowledge from past tasks gets transferred to the new task by LiZero, it outperforms all baselines with more significantly improved convergence speed. In Task 10 with maximum past knowledge, LiZero demonstrates the largest improvement in convergence speed and optimal reward.

| Name | Task1 | Task2 | Task3 | Task4 | Task5 | Task6 | Task7 | Task8 | Task9 | Task10 | Total |
|------|-------|-------|-------|-------|-------|-------|-------|-------|-------|--------|-------|
| LiZero-U | 4.83±0.05 | 5.98±0.10 | 5.99±0.07 | 5.94±0.05 | 6.08±0.07 | 6.05±0.16 | 6.01±0.11 | 6.03±0.05 | 6.04±0.04 | 6.03±0.09 | 58.98 |
| LiZero-P | 4.65±0.06 | 5.89±0.11 | 5.90±0.12 | 5.90±0.03 | 5.62±0.19 | 5.68±0.22 | 5.76±0.12 | 5.87±0.03 | 5.78±0.06 | 5.79±0.20 | 56.84 |
| LiZero-N | 4.64±0.08 | 5.56±0.07 | 5.56±0.07 | 5.52±0.05 | 5.52±0.08 | 5.48±0.06 | 5.50±0.09 | 5.45±0.06 | 5.50±0.06 | 5.48±0.05 | 54.21 |
| MCTS-R | 4.51±0.07 | 4.43±0.08 | 4.32±0.11 | 4.24±0.05 | 4.18±0.07 | 4.24±0.10 | 4.25±0.03 | 4.47±0.06 | 4.34±0.03 | 4.39±0.08 | 43.37 |
| MCTS-O | 4.52±0.06 | 4.87±0.08 | 4.57±0.04 | 4.16±0.03 | 4.78±0.05 | 4.91±0.06 | 4.04±0.03 | 3.70±0.05 | 3.02±0.07 | 2.96±0.06 | 41.53 |
| pUCT | 4.66±0.04 | 4.71±0.06 | 4.69±0.13 | 4.77±0.09 | 4.74±0.04 | 4.87±0.05 | 4.94±0.06 | 4.72±0.05 | 4.86±0.07 | 4.77±0.03 | 47.73 |
| RMax | 1.02±0.02 | 1.05±0.01 | 1.01±0.02 | 1.03±0.01 | 1.04±0.01 | 1.05±0.01 | 1.03±0.03 | 1.04±0.02 | 1.03±0.02 | 1.03±0.01 | 10.33 |
| LRMax | 1.05±0.01 | 1.05±0.02 | 1.04±0.02 | 1.06±0.03 | 1.05±0.01 | 1.06±0.02 | 1.04±0.01 | 1.06±0.03 | 1.05±0.01 | 1.04±0.01 | 10.50 |

Table 1: The table summarizes the rewards and standard deviations obtained in sequential tasks. It shows that LiZero achieves about 31% early reward improvement on average, compared with MCTS baselines (including two versions of MCTS with UCT Winands (2024); Kocsis & Szepesvári (2006); Cheng et al. and one with pUCT similar to MuZero Schrittwieser et al. (2020a)) and lifelong RL baselines (including RMax Brafman & Tennenholtz (2002) and LRMax Lecarpentier et al. (2021a)). MCTS-R and MCTS-O demonstrate similar level of performance, both better than lifelong RL and slightly below pUCT. LiZero algorithms outperform MCTS baselines by about 31% early reward improvement on average. With more accurate distance estimates – i.e., from Lizero-N to LiZero-P and LiZerio-U – we observe further improvement due to better knowledge transfer that comes with more accurate aUCT.

In the evaluation, we consider some state-of-the-art baselines using MCTS and lifelong RL. In particular, we consider two versions of MCTS algorithms that leverage UCT Winands (2024); Kocsis & Szepesvári (2006); Cheng et al.: MCTS-R denotes a version that restarts the search from scratch for each new task, and MCTS-O denotes a version that is oblivious to the sequence of distinct task dynamics and continues to build upon the search tree from the past. We also consider state-of-the-art MCTS using pUCT, similar to MuZero and related algorithms Schrittwieser et al. (2020a). We have two lifelong RL algorithms: RMax Brafman & Tennenholtz (2002) and LRMax Lecarpentier et al. (2021a), which exploits a similar Lipschitz

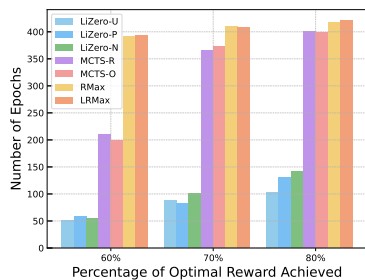

Figure 2: LiZero shows a comfortable speedup of 3∼4x, compared with MCTS and lifelong RL baselines, to achieve the same level of optimal rewards with higher sample efficiency.

continuity in RL but does not consider MCTS using upper confidence bounds. We evaluated three versions of LiZero using different methods to estimate aUCT by computing task distances, as presented in Section 4. LiZero-U employs a direct distance estimate based on Definition 3.1; LiZero-P is the data-driven distance estimater $\hat{d}_2$ using samples following the search policy; and LiZero-N is the neural-network based estimator $\hat{d}_{para}$ using parameter distances.

The experimental environment we used is a variation of the "tight" task by Abel et al. Abel et al. (2018). It generates a sequence of ten learning tasks. Each task consists of a $25 \times 25$ grid world, with the initial state located at the center, and four possible actions: up, down, left, and right. The three

cells in the top-right corner and one cell in the bottom-left corner are designated as goal cells. For each task, the reward for the goal cells is randomly chosen from the range [0.9, 1]. The remaining cells will randomly generate interference rewards within the range [0, 0.1]. Its state transition matrix selects its own slip probability (acting differently from the chosen one) within the range [0, 0.1]. This ensures that the sequence of tasks has varying reward and transition probabilities. Each task is for 1,000 epochs. These operations are repeated multiple times to narrow the confidence interval.

Figure 1 shows the convergence of different algorithms on representative Tasks 1, 2, 6, and 10, in a sequence of ten distinct tasks. As tasks are drawn sequentially, LiZero-U, LiZero-P, and LiZero-N algorithms converge more rapidly than the MCTS and lifelong RL baselines. This speedup becomes evident as early as the second task (Task 2) – while similar convergences are observed in Task 1 as no prior knowledge is yet available. From Task 2 to Task 10, as more knowledge from past tasks gets transferred to the new task by LiZero, it outperforms all baselines in significantly more improved convergence speed. In Task 10 with maximum past knowledge, LiZero outperforms all baselines in convergence speed and optimal reward. MCTS-O (which is oblivious to changing task dynamics) exhibits worse performance than MCTS-R (which restarts from scratch).

In Table 1, we summarize the average rewards (and their standard deviations) obtained in sequential tasks by different algorithms during their first 500 epochs (i.e., first half of the learning process). LiZero algorithms achieves about 31% early reward improvement on average. As for MCTS baselines with UCT, MCTS-R shows similar reward across different tasks, while MCTS-O demonstrates higher volatility – due to its reliance on how task dynamics evolve. pUCT achieves higher performance due to the use of improved probabilistic UCT similar to MuZero. All MCTS baselines show better results than lifelong RL algorithms (i.e., RMax and LRMax), which are known to be less sample efficient and require more epochs for exploration/exploitation. With more accurate distance estimates – i.e., from Lizero-N to LiZero-P and to LiZero-U – we observe further improved results due to better knowledge transfer that comes with more accurate aUCT calculations.

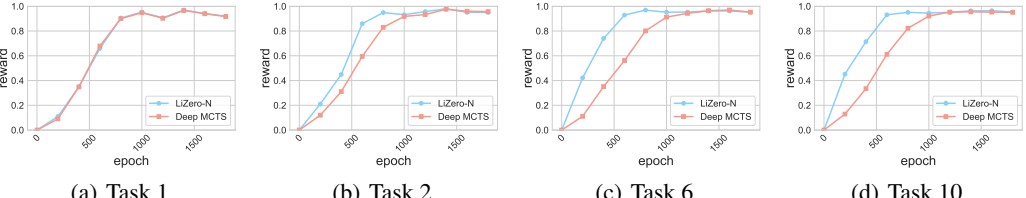

|         |         |         |          |
|---------|---------|---------|----------|
| (a) Task 1 | (b) Task 2 | (c) Task 6 | (d) Task 10 |

Figure 3: Performance of LiZero-N and Deep MCTS on a continuous-state MuJoCo Point Maze sequence.Each panel shows the normalized return versus planning epochs for Tasks 1, 2, 6, and 10.On the first task LiZero-N performs similarly to Deep MCTS, while on later tasks it converges faster and achieves higher early returns, demonstrating that aUCT-based transfer remains effective in continuous MDPs and that its benefits grow as more tasks are learned.

To demonstrate that the proposed aUCT transfer is not limited to discrete gridworlds, we further evaluate LiZero-N in a continuous state-space environment, the MuJoCo Point Maze. The environment consists of a point mass with two degrees of freedom $(x, y)$ that is force-actuated in Cartesian coordinates and must reach a target goal position inside a closed maze. We construct a sequence of 10 tasks by randomly modifying the maze parameters and goal locations. Each task is solved by running MCTS for 2,000 planning epochs. We compare LiZero-N with the same Deep MCTS baseline used in the gridworld experiments. Figure 3 reports the normalized returns averaged over multiple runs.

To evaluate the speedup of LiZero, Figure 2 shows the average number of epochs needed by different algorithms to achieve 60%, 70%, and 80% of the optimal reward, respectively. We note that LiZero shows a comfortable speedup of 3∼4x, compared to MCTS and lifelong RL baselines, while RL baselines are much less sample-efficient than MCTS-based planning, in general. We do not go beyond 80% in this plot since some baselines are never able to achieve more than 80% of the optimal reward that LiZero obtains. The results validate the acceleration as characterized by $\Gamma$ in Theorem 3.4.

**Ablation Study.** Our ablation study considers the impact of distance estimator on performance. Figure 4 shows the distance estimators in LiZero-U, LiZero-P, and LiZero-N (each with decreasing accuracy) across the sequence of tasks, while for the purpose of ablation study, MCTS-R can be viewed as an algorithm without distance estimator. Comparing the performance of these algorithms in

Table 1 and Figure 2, we see that the superior performance of LiZero is indeed resulted from the use of aUCT in MCTS – The tighter aUCT bounds we use, the higher performance we can achieve. Using no distance estimator and thus only UCT (in MCTS-R) leads to the lowest performance. Further, as tasks are drawn, the distance estimates decrease quickly, and by the third task, it is already very small, implying accurate aUCT calculation for knowledge transfer.

## 6 CONCLUSIONS

We study theoretically the transfer of past knowledge in MCTS-based lifelong planning and develop a novel aUCT rule, depending on both Lipschitz continuity between tasks and the confidence of knowledge in Monte Carlo action sampling. The proposed aUCT is proven to significantly accelerate MCTS and enable a new lifelong MCTS algorithm: LiZero. We present efficient methods for online estimation of aUCT and analyze the sampling complexity and error bounds. LiZero is evaluated on a sequence of distinct tasks with varying transition probabilities and rewards. It outperforms MCTS and lifelong RL baselines with 3~4x speed-up and about 31% higher early reward.

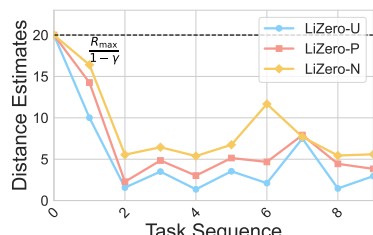

Figure 4: Our ablation study comparing different distance estimators in LiZero-U, LiZero-P, and LiZero-N, while MCTS-R can be viewed as a baseline without distance estimator. The relevant performance of these algorithms are provided in Table 1 and Figure 2 and thus not repeated here. The superior performance of LiZero is indeed resulted from the use of aUCT in MCTS. The tighter aUCT bounds, the higher performance we can obtain.

**Limitations:** The derivation of Theorem 3.2 considers varying transition probabilities and rewards, while assuming the same state and action spaces. In this case, we could consider the union of state and action spaces, but better approaches may be needed. Further, our acceleration analysis requires that the optimal Q-function has a bounded *advantage gap* to compute the acceleration factor $\Gamma$. This bound may be loose for practical problems, leading to underestimation of acceleration factor $\Gamma$.

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

# A    APPENDIX / SUPPLEMENTAL MATERIAL

## THE USE OF LARGE LANGUAGE MODELS (LLMS)

The authors did not use Large Language Models for research ideation, derivations, proofs, experimental design, data analysis, or writing of the manuscript. No LLM contributed content that would qualify as authorship or a significant contribution under the conference policy.

## IMPACT STATEMENT

This paper proposes a novel framework for applying Monte Carlo Tree Search (MCTS) in lifelong learning settings, addressing the challenges posed by non-stationary environments and dynamic game dynamics. By introducing the adaptive Upper Confidence Bound for Trees (aUCT) and leveraging insights from previous MDPs (Markov Decision Processes), our work significantly enhances the efficiency and adaptability of decision-making algorithms across evolving tasks.

The broader societal implications of this research include its potential to improve AI applications in robotics, automated systems, and other domains requiring dynamic decision-making under uncertainty. For instance, this framework could be used in autonomous systems to adaptively respond to changing environments, thereby improving safety and reliability. At the same time, it is crucial to acknowledge and mitigate potential risks, such as unintended biases or over-reliance on prior knowledge that may not fully represent novel situations.

Ethical considerations for this work focus on its use in high-stakes applications, such as healthcare, finance, or defense, where decision-making under uncertainty could have significant consequences. Developers and practitioners should implement safeguards to ensure responsible deployment, including comprehensive testing in diverse scenarios and establishing clear boundaries for its use.

By advancing the state of the art in continual learning and decision-making, this research contributes to the development of more adaptable and intelligent AI systems while highlighting the importance of ethical and responsible innovation in AI technologies.

### A.1    PROOF OF THEOREM 3.2

*Proof.* Proof of Theorem 3.2 Since in the MCTS UCB algorithm, the estimated Q-values are obtained through multiple simulations, we need to analyze how the differences in simulation results between two MDPs affect the estimated Q-values.

However, due to the randomness involved in the simulation process of the two MDPs:

- **Transition randomness:** Due to different transition probabilities, the two MDPs may move to different next states even when starting from the same state and action.

- **Action selection randomness:** When using the UCB algorithm, action selection depends on the current statistical information, which in turn relies on the past simulation results.

The randomness mentioned above makes it impossible for us to compare two independent random simulation processes directly.

To eliminate the impact of randomness, we need to construct a coupled simulation process for the two MDPs in the same probability space, allowing for a direct comparison between them. Then we will incorporate the additional errors caused by randomness into the analysis as error terms. For this purpose, we present the following assumptions.

**Assumption A.1.** Let us temporarily assume that the actions selected in each simulation are the same for the two MDPs.

- **Initial action consistency:** The simulation starts from the same state $s$

- **Action selection consistency:** The same action $a$ is chosen in each state.

Note: This is a strong assumption and may not hold in practice. We will discuss its impact later.

Thus, we can obtain the difference in cumulative rewards between the two MDPs in a single simulation as:

$$\Delta G = G_M - G_{M'} = \sum_{t=0}^{T} \gamma^t (R(s_t^M, a_t) - R'(s_t^{M'}, a_t)) \tag{10}$$

Where $s_t^M$ and $s_t^{M'}$ are the states of the two MDPs at step $t$, and $a_t$ is the action selected at step $t$.

So we can get

$$\left| Q_M^N(s,a) - Q_{M'}^{N'}(s,a) \right| = \left| \frac{1}{N} \sum_{i=1}^{N} G_{M,i} - \frac{1}{N'} \sum_{i=1}^{N'} G_{M,i} \right| \leq \bar{\Delta G} = \left| \frac{1}{n} \sum_{i=1}^{n} \Delta G_i \right| \tag{11}$$

where $n = \min\{N, N'\}$ To estimate the expectation and variance of $\Delta G$, we need to analyze how the differences in the state sequences affect the cumulative rewards.

We present several settings for the state differences.

- **Probability of state difference:** At each time step $t$, the probability that the states of the two MDPs differ is denoted as $p_t$.

- **Initial state is the same:** $p_0 = 0$.

- **State difference propagation:** Due to differences in transition probabilities, state differences may accumulate in subsequent time steps.

Since the probability of state differences occurring at each step is difficult to calculate precisely, we can use the total variation distance to estimate the probability of transitioning to different states. We present the definition of the total variation distance between the transition probabilities of the two MDPs and a recursive method for calculating the probability of state differences.

**Definition A.2.** Under action $a_t$, starting from state $s_t$, the total variation distance between the transition probabilities of the two MDPs is:

$$D_{TV}(P, P') = \frac{1}{2} \sum_{s'} |P(s'|s_t, a_t) - P'(s'|s_t, a_t)| \tag{12}$$

Thus, starting from the same state $s_t$ and action $a_t$, the probability that the two MDPs transition to different next states is at most $D_{TV}(P, P') \leq \frac{\Delta P}{2}$.

Thus, the probability of state differences occurring can be recursively expressed as:

$$p_{t+1} \leq p_t + (1 - p_t) \cdot D_{TV}(P, P') \leq p_t + \frac{\Delta P}{2} \tag{13}$$

So

$$p_t \leq t \cdot \frac{\Delta P}{2} \tag{14}$$

Thus, at each time step $t$, the expected difference in cumulative rewards is:

$$\mathbb{E}[|\Delta G|] = \mathbb{E}[\sum_{t=0}^{T} \gamma^t (R(s_t^M, a_t) - R'(s_t^{M'}, a_t))]$$

$$= \sum_{t=0}^{T} \gamma^t ( \underbrace{\mathbb{E}[R(s_t^M, a_t) - R'(s_t^M, a_t)]}_{\text{The impact of reward function differences}} + \underbrace{\mathbb{E}[R'(s_t^M, a_t) - R'(s_t^{M'}, a_t)]}_{\text{Reward differences caused by state differences}} )$$

$$\leq \sum_{t=0}^{T} \gamma^t (\Delta R + 2R_{\max} \cdot p_t) \tag{15}$$

$$= \frac{\Delta R}{1 - \gamma} + \sum_{t=0}^{T} \gamma^t \cdot 2R_{\max} \cdot t \cdot \frac{\Delta P}{2}$$

$$= \frac{\Delta R}{1 - \gamma} + R_{\max}\Delta P \sum_{t=0}^{T} t\gamma^t$$

$$= \frac{\Delta R}{1 - \gamma} + R_{\max}\Delta P \cdot \frac{\gamma}{(1 - \gamma)^2}$$

To estimate the variance of the cumulative reward difference, since the cumulative reward is bounded, its variance is also finite. We can easily obtain

$$|\Delta G| \leq G_{\max} = \frac{2R_{\max}}{1 - \gamma} \tag{16}$$

According to Hoeffding:

$$P(|\bar{\Delta G} - \mathbb{E}[\bar{\Delta G}]| \geq \epsilon) \leq 2\exp(-\frac{2n\epsilon^2}{G_{\max}^2}) \tag{17}$$

Thus, with probability at least $1 - \delta$, we have:

$$|\hat{Q}_M^n(s, a) - \hat{Q}_{M'}^n(s, a)| \leq \mathbb{E}[|\Delta \bar{G}|] + G_{\max}\sqrt{\frac{\ln(2/\delta)}{2n}}$$

$$= \frac{\Delta R}{1 - \gamma} + R_{\max}\Delta P \cdot \frac{\gamma}{(1 - \gamma)^2} + \frac{2R_{\max}}{1 - \gamma}\sqrt{\frac{\ln(2/\delta)}{2n}} \tag{18}$$

$$= \frac{1}{1 - \gamma}(\Delta R + \frac{R_{\max}\gamma}{1 - \gamma}\Delta P) + \frac{2R_{\max}}{1 - \gamma}\sqrt{\frac{\ln(2/\delta)}{2n}}$$

$$= L(\Delta R + \kappa\Delta P) + L_2$$

$$\square$$

## A.2    PROOF OF THEOREM 3.4

*Proof.* Proof of Theorem 3.4 First, we consider the case of a single MDP and assume that we have a "universal" upper bound $U(s, a) \geq Q_M^*(s, a)$.

**Lemma A.3.** *Since $U(s, a) \geq Q_M^*$ holds for all $(s, a)$, and initially $Q(s, a) \leq U(s, a)$, for any update, $Q(s, a)$ maintains $Q(s, a) \leq U(s, a)$ and $Q(s, a) \geq$ (a non-negative expected estimate).*

The above two points illustrate Since we update using $Q(s, a) = \min\{\hat{Q}(s, a), U(s, a)\}$ And since $U(s, a) \geq Q^*(s, a)$, during all sampling processes, if $\hat{Q}(s, a)$ overestimates $Q^*(s, a)$ significantly, it will still be truncated by $U(s, a)$, ensuring that $Q(s, a) \leq U(s, a)$. When $\hat{Q}(s, a)$ gradually approaches $Q^*(s, a)$, it will no longer be truncated. This does not hinder the convergence of $Q$ to $Q^*$.

**Theorem A.4** (Convergence in a Single MDP). *If there are infinitely many samples for each state $s$ and its available actions $a$ (i.e., every branch in the MCTS search tree is "continuously" expanded), then the $Q(s, a)$ generated by the above update formula almost surely converges to $Q_M^*(s, a)$.*

Now we aim to demonstrate that after completing certain MDPs (tasks) $\bar{M}_1, \bar{M}_2, \ldots, \bar{M}_m$, and then switching to a new MDP $M$, the algorithm achieves faster convergence.

First, we analyze the classic scenario without upper bounds. In a finite state-action space, to achieve the desired outcome with high probability $1 - \delta$:

$$\max_{(s,a) \in \mathcal{S} \times \mathcal{A}} |Q_n(s, a) - Q_M^*(s, a)| \leq \epsilon \tag{19}$$

The standard UCT/UCB theory typically provides a time complexity of $\tilde{O}\left(\frac{|\mathcal{S}||\mathcal{A}|}{(1-\gamma)^3 \epsilon^2} \ln \frac{1}{\delta}\right)$. To prove this theorem, we just need to analyze the acceleration factor $\Gamma$, comparing the sampling complexity of our aUCT and standard UCT.

More specifically, if we examine each specific $(s, a)$, the analysis often resembles that of multi-armed bandits: for "suboptimal" $(s, a)$, approximately $\tilde{O}\left(\frac{1}{(\Delta_{(s,a)}^M)^2} \ln \frac{1}{\delta}\right)$ samples are required. Where $\Delta_{(s,a)}^M = Q_M^*(s, a^*) - Q_M^*(s, a)$ is the value gap between the action and the optimal action. Summing up the exploration costs for all state-action pairs gives a total magnitude of $\sum_{(s,a)} \frac{1}{(\Delta_{(s,a)}^M)^2}$.

Now we introduce the case with upper bounds and analyze how to reduce the number of samples across different MDPs.

To quantitatively represent this acceleration, we divide the state-action pairs $(s, a)$ into two groups:

- $\mathcal{S}_1$ : Upper bounds are sufficiently tight and are truncated to be lower than the optimal action from the very beginning.

$$\mathcal{S}_1 = \left\{(s, a) | \exists i : U_{\bar{M}_i}(s, a) < Q_M^0(s, a)\right\} \tag{20}$$

- $\mathcal{S}_0$ : The upper bounds are not "tight enough," i.e.,

$$\mathcal{S}_0 = \text{remaining actions} \tag{21}$$

For $(s, a) \in \mathcal{S}_1$:

We treat each sampling as a multi-armed bandit. Let the true mean of the optimal arm be $\mu^*$. For a certain arm $j$, its true mean is known to satisfy $\mu_j \leq U_j < \mu^*$.

Even if we truncate $\hat{\mu}_n(j)$ at $U_j$, the UCB algorithm's "optimistic estimate" for this arm at step $n$ is still:

$$Q_n(j) = \min\{\hat{\mu}_n(j), U_j\} + c\sqrt{\frac{\ln(n)}{N_j(n)}} \tag{22}$$

$$U_j + c\sqrt{\frac{\ln(n)}{N_j(n)}} < \mu^* \tag{23}$$

Let $\Delta = \mu^* - U_j$. As long as:

$$\sqrt{\frac{\ln(n)}{N_j(n)}} \leq \frac{\Delta}{2c} \tag{24}$$

From the above, it can be ensured that $Q_n(j)$ cannot exceed $\mu^* - \Delta/2$. So

$$N_j(n) \geq \frac{4c^2 \ln(n)}{\Delta^2} \tag{25}$$

Where we obtain a sampling time complexity of $\tilde{O}(\ln n)$.

For $(s,a) \in \mathcal{S}_0$, these $(s,a)$ cannot be pruned by "truncation." They still require multiple samples, as in classic UCT, to determine whether they are truly optimal. For any $(s,a) \in \mathcal{S}_0$, we still need approximately $O\left(\frac{1}{(\Delta_{(s,a)}^M)^2} \ln \frac{1}{\delta}\right)$ samples to distinguish that it is not as good as $(s,a^*)$. Thus, the sampling complexity of our algorithm is:

$$X_{\mathrm{aUCT}} = \sum_{(s,a) \in \mathcal{S}_1} \tilde{O}\left(\ln n\right) + \sum_{(s,a) \in \mathcal{S}_0} \tilde{O}\left(\frac{1}{(\Delta_{(s,a)}^M)^2} \ln \frac{1}{\delta}\right), \tag{26}$$

Using the fact that $\tilde{O}(\ln n) \sim \tilde{O}(\ln \frac{1}{\delta})$, we can rewrite this as

$$X_{\mathrm{aUCT}} = \sum_{(s,a) \in \mathcal{S}_1} \tilde{O}\left(\ln \frac{1}{\delta}\right) + \sum_{(s,a) \in \mathcal{S}_0} \tilde{O}\left(\frac{1}{(\Delta_{(s,a)}^M)^2} \ln \frac{1}{\delta}\right). \tag{27}$$

In contrast, the sampling complexity of the standard UCT can be obtained using the same analysis, i.e.,

$$X_{\mathrm{UCT}} = \sum_{(s,a) \in \mathcal{S}_0 \cup \mathcal{S}_1} \tilde{O}\left(\frac{1}{(\Delta_{(s,a)}^M)^2} \ln \frac{1}{\delta}\right). \tag{28}$$

Comparing the order bounds from Equation (28) and Equation (27), we can find the acceleration factor $\Gamma$ as

$$\Gamma = \frac{\sum_{(s,a) \in \mathcal{S}_1 \cup \mathcal{S}_0} \frac{1}{(\Delta_{(s,a)}^M)^2}}{\sum_{(s,a) \in \mathcal{S}_1} (1) + \sum_{(s,a) \in \mathcal{S}_0} \frac{1}{(\Delta_{(s,a)}^M)^2}}, \tag{29}$$

which is the desired result in the theorem.

$\square$

## A.3 PROOF OF THEOREM 4.1

*Proof.* Proof of Theorem 4.1 First, we need to establish unbiasedness and boundedness. For unbiasedness, we can derive:

$$\mathbb{E}[X_i] = \mathbb{E}_{(s,a) \sim \pi}\left[\frac{\mathcal{U}(s,a)}{\pi(s,a)} \cdot \Delta X(s,a)\right] = \mathbb{E}_{(s,a) \sim \mathcal{U}}[\Delta X(s,a)] = d(M, M') \tag{30}$$

Therefore, $\mathbb{E}[\hat{d}_{\mathcal{U}}] = d(M, M')$, meaning $\hat{d}_{\mathcal{U}}$ is an unbiased estimator.

$$w_i = \frac{\mathcal{U}(s_i, a_i)}{\pi(s_i, a_i)} \leq \frac{\mathcal{U}_{\max}}{\alpha} \tag{31}$$

Where $\mathcal{U}\max = \max_{(s,a)} \mathcal{U}(s,a) = \frac{1}{|\mathcal{S}| \cdot |\mathcal{A}|}$. So we can get:

$$X_i = w_i \Delta X(s_i, a_i) \leq \left(\frac{\mathcal{U}_{\max}}{\alpha}\right) b \tag{32}$$

So we can get $X_i \in [0, C]$ where $C = \frac{\mathcal{U}_{\max}}{\alpha} b$.

Based on the above analysis, we have $\bar{X}_N = \frac{1}{N} \sum_{i=1}^N X_i = \hat{d}_{\mathcal{U}}, \mu = \mathbb{E}[X_i] = d(M, M')$. According to Hoeffding's inequality, for $\bar{X}_N \in [0, C]$, we have:

$$\Pr\{|\bar{X}_N - \mu| \geq \epsilon\} \leq 2 \exp\left(-\frac{2N\epsilon^2}{C^2}\right) \tag{33}$$

To achieve a confidence level of $\delta$, it requires:

$$2 \exp\left(-\frac{2N\epsilon^2}{C^2}\right) \leq \delta \Leftrightarrow \exp\left(-\frac{2N\epsilon^2}{C^2}\right) \leq \frac{\delta}{2} \Leftrightarrow -\frac{2N\epsilon^2}{C^2} \leq \ln \frac{\delta}{2} \Leftrightarrow \frac{2N\epsilon^2}{C^2} \geq \ln \frac{2}{\delta} \Leftrightarrow N \geq \frac{C^2}{2\epsilon^2} \ln \frac{2}{\delta} \tag{34}$$

We get if fulfilled:

$$N \geq \frac{1}{2\epsilon^2}(\frac{\mathcal{U}_{\max}}{\alpha}b)^2 \ln\frac{2}{\delta} \tag{35}$$

There is then a high probability error upper bound:

$$\Pr\{|\hat{d}_{\mathcal{U}} - d(M, M')| \leq \epsilon\} \geq 1 - \delta \tag{36}$$

□

### A.4 PROOF OF THEOREM 4.2

*Proof.* Proof of Theorem 4.2 Constructing a martingale difference, let:

$$S_n := \sum_{k=1}^{n}(X_k - d(M, M')), Y_k := X_k - \mathbb{E}[X_k|\mathcal{F}_{k-1}] \tag{37}$$

According to the martingale condition in formula 9, we know that $Y_k = X_k - d(M, M')$, and $S_n = \sum_{k=1}^{n} Y_k$ satisfies $\mathbb{E}[Y_k|\mathcal{F}_{k-1}] = 0$. Thus, $\{S_n, \mathcal{F}_n\}$ is a martingale process.

Since $\pi_k(s, a) \geq \alpha \Rightarrow w_k \leq \frac{\mathcal{U}_{\max}}{\alpha}$, and $\Delta X(s, a) \leq b \Rightarrow X_k = w_k \Delta X(s_k, a_k) \leq \frac{\mathcal{U}_{\max}}{\alpha}b =: C$. Therefore, we have:

$$|Y_k| \leq \max\{X_k, d(M, M')\} \leq C \tag{38}$$

According to the Azuma-Hoeffding inequality for bounded martingale differences, we have:

$$\Pr\{|S_n| \geq t\} \leq 2\exp(-\frac{t^2}{2NC^2}) \tag{39}$$

Let $t = N\epsilon$, then $|S_n| \geq t$ is equivalent to $|\sum_{k=1}^{n} X_k - Nd(M, M')| \geq N\epsilon$, that is:

$$|\hat{d}_{\mathcal{U}}^{(n)} - d(M, M')| \geq \epsilon \tag{40}$$

So:

$$\Pr\{|\hat{d}_{\mathcal{U}}^{(N)} - d(M, M')| \geq \epsilon\} \leq 2\exp(-\frac{N\epsilon^2}{2C^2}) \tag{41}$$

Thus, as long as $N \geq \frac{2C^2}{\epsilon^2}\ln\frac{2}{\delta}$, we have $\Pr\{|\hat{d}_{\mathcal{U}}^{(N)} - d(M, M')| \geq \epsilon\} \leq \delta$. □

### A.5 PROOF OF THEOREM 4.3

*Proof.* Proof of Theorem 4.3 We decompose $d_{\mathcal{U}}$.

$$\begin{aligned}
d_{\mathcal{U}}(M, M_i) &= \mathbb{E}_{(s,a)\sim\mathcal{U}}[\underbrace{|R_s^a - R_s^{a,(i)}|}_{\text{Reward difference}} + \kappa\underbrace{\sum_{s'}|P_{ss'}^a - P_{ss'}^{a,(i)}|}_{\text{transition difference}}] \\
&\simeq \mathbb{E}_{(s,a)\sim\mathcal{U}}[|R_s^a - R_s^{a,(i)}| + \kappa||\Psi_\phi(s, a) - \Psi_{\phi_i}(s, a)||_1] \\
&\leq \mathbb{E}_{(s,a)\sim\mathcal{U}}[L_3\rho(\phi, \phi_i) + \kappa L_3\rho(\phi, \phi_i)] \\
&\leq L_3\rho(\phi, \phi_i) + \kappa L_3\rho(\phi, \phi_i) \\
&= (1 + \kappa)L_3\rho(\phi, \phi_i) \\
&= (1 + \kappa)L_3\hat{d}_{para}(M, M_i)
\end{aligned} \tag{42}$$

□

## B PSEUDO-CODE

---

**Algorithm 1** UMCTS

---

**Require:** $\{\mathcal{M}_1, \ldots, \mathcal{M}_M\}, \mathcal{U}, \kappa, L, L_2^{(i)}, \gamma, R_{\max}, C, T$

1: **for** $i = 1$ to $M$ **do**
2:    Repeat sampling $(s, a)$ from the uniform distribution $\mathcal{U}$ to update $R$ and $P$.
3:    **for** $j = 1$ to $M$ **do**
4:       $d(\mathcal{M}_i, \mathcal{M}_j) \leftarrow \mathbb{E}_{(s,a,s') \sim \mathcal{U}} \big[\, |R_s^a - \overline{R}_s^a| + \kappa\, |P_{ss'}^a - \overline{P}_{ss'}^a| \,\big]$
5:    **end for**

6:    Initialize root node $s_0$, set $N(\cdot), N(\cdot, \cdot), W(\cdot, \cdot)$ to 0
7:    **for** $t = 1$ to $T$ **do**
8:       **Selection**:
9:         Set current node $s \leftarrow s_0$
10:      **while** child nodes of $s$ are fully expanded **do**
11:        Choose $a = \arg\max_a \big(Q(s, a)\big)$    // using Eq. (*) below
12:        $s \leftarrow$ child node after action $a$
13:      **end while**
14:      **Expansion**:
15:        Expand one non-visited action $a_{\text{new}}$ at $s$, sample $s'$ from environment or model
16:        Create new child node $s'$, set $N(s', \cdot) = 0, W(s', \cdot) = 0$
17:      **Simulation**:
18:        Perform a (light) rollout or default policy from $s'$ to terminal or horizon
19:        Receive cumulative reward $G$
20:      **Backpropagation**:
21:        Traverse back from $s'$ to $s_0$ along visited path
22:      **for all** visited state-action pairs $(\tilde{s}, \tilde{a})$ **do**
23:        $N(\tilde{s}) \leftarrow N(\tilde{s}) + 1$
24:        $N(\tilde{s}, \tilde{a}) \leftarrow N(\tilde{s}, \tilde{a}) + 1$
25:        $W(\tilde{s}, \tilde{a}) \leftarrow W(\tilde{s}, \tilde{a}) + G$
26:        // Update $Q(\tilde{s}, \tilde{a})$ with UMCTS bound:
27:        $U_{\bar{\mathcal{M}}}(\tilde{s}, \tilde{a}) \leftarrow Q_{\bar{\mathcal{M}}}^*(\tilde{s}, \tilde{a}) + L \cdot d(\mathcal{M}, \bar{\mathcal{M}}) + L_2^{(i)}$
28:        $U(\tilde{s}, \tilde{a}) \leftarrow \min\big\{ \frac{R_{\max}}{1-\gamma}, U_{\bar{\mathcal{M}}}(\tilde{s}, \tilde{a}), \ldots \big\}$
29:        $Q(\tilde{s}, \tilde{a}) \leftarrow \min\Big\{ \dfrac{W(\tilde{s}, \tilde{a})}{N(\tilde{s}, \tilde{a})} + C\sqrt{\dfrac{\ln N(\tilde{s})}{N(\tilde{s}, \tilde{a})}}, U(\tilde{s}, \tilde{a}) \Big\}$   (*)
30:      **end for**
31:    **end for**
32: **end for**

---

---

**Algorithm 2** UMCTS with Importance Sampling

---

**Require:** Tasks $\{\mathcal{M}_1, \ldots, \mathcal{M}_M\}$, each partially known; Uniform distribution $\mathcal{U}(s, a)$;Lipschitz constants $L$, $L_2^{(i)}$; Discount factor $\gamma$, maximum reward $R_{\max}$; Exploration constant $C$; Number of search iterations $T$;A (default) policy $\pi$ used in Simulation for importance sampling;

1: **Function** DISTANCE$(\mathcal{M}, \bar{\mathcal{M}}, \pi)$:

2:     $\Delta X(s, a) \triangleq \Delta R_s^a + \kappa \, \Delta P_s^a$

3:     **return** $\mathbb{E}_{(s,a)\sim\pi} \left[ \dfrac{\mathcal{U}(s, a)}{\pi(s, a)} \cdot \Delta X(s, a) \right]$

4: // **For each task** $\mathcal{M}_i$

5: **for** $i = 1$ to $M$ **do**

6:     Initialize root node $s_0$, set $N(\cdot) = 0$, $N(\cdot, \cdot) = 0$, $W(\cdot, \cdot) = 0$

7:     (Optionally maintain a buffer $\mathcal{D}_i$ for storing samples (s,a))

8:     **for** $t = 1$ to $T$ **do**

9:         **Selection:**

10:         $s \leftarrow s_0$

11:         **while** all actions from $s$ are fully expanded **and** $s$ not terminal **do**

12:             $a \leftarrow \underset{a}{\operatorname{argmax}} \left( Q(s, a) \right)$     // UCB or UMCTS criterion

13:             $s \leftarrow$ child node after action $a$

14:         **end while**

15:         **Expansion:**

16:         **if** $s$ not terminal **then**

17:             Choose one unvisited action $a_{\text{new}}$ at $s$

18:             Sample next state $s' \sim P_i(\cdot \mid s, a_{\text{new}})$ // from environment or model

19:             Create child node $s'$, set $N(s', \cdot) = 0$, $W(s', \cdot) = 0$

20:         **end if**

21:         **Simulation:**

22:         Initialize cumulative reward $G \leftarrow 0$

23:         $s_{\text{sim}} \leftarrow s'$

24:         **while** $s_{\text{sim}}$ is not terminal **do**

25:             Pick action $a_{\text{sim}}$ by policy $\pi(\cdot \mid s_{\text{sim}})$

26:             Observe reward $r_{\text{sim}} = R_i(s_{\text{sim}}, a_{\text{sim}})$

27:             Observe next state $s_{\text{next}} \sim P_i(\cdot \mid s_{\text{sim}}, a_{\text{sim}})$

28:             $G \leftarrow G + r_{\text{sim}}$

29:             // Update or record increments for $R_s^a$, $P_{s,s'}^a$

30:               $\Delta R_{s_{\text{sim}}}^a$, $\Delta P_{s_{\text{sim}}}^a \leftarrow$ (computed from new sample)

31:             // Optionally store $(s_{\text{sim}}, a_{\text{sim}})$ in $\mathcal{D}_i$ for importance sampling

32:             $s_{\text{sim}} \leftarrow s_{\text{next}}$

33:         **end while**

34:         **Backpropagation:**

35:         Traverse from $s'$ back to $s_0$ along visited path

36:         **for all** visited pairs $(\tilde{s}, \tilde{a})$ **do**

37:             $N(\tilde{s}) \leftarrow N(\tilde{s}) + 1$

38:             $N(\tilde{s}, \tilde{a}) \leftarrow N(\tilde{s}, \tilde{a}) + 1$

39:             $W(\tilde{s}, \tilde{a}) \leftarrow W(\tilde{s}, \tilde{a}) + G$

40:             /* Use the Lipschitz bound with distance estimation */

41:             $d(\mathcal{M}_i, \bar{\mathcal{M}}) \leftarrow \text{Distance}(\mathcal{M}_i, \bar{\mathcal{M}}, \pi)$

42:             $U_{\bar{\mathcal{M}}}(\tilde{s}, \tilde{a}) \leftarrow Q_{\bar{\mathcal{M}}}^*(\tilde{s}, \tilde{a}) + L \cdot d(\mathcal{M}_i, \bar{\mathcal{M}}) + L_2^{(i)}$

43:             $U(\tilde{s}, \tilde{a}) \leftarrow \min\left\{ \dfrac{R_{\max}}{1 - \gamma}, U_{\bar{\mathcal{M}}}(\tilde{s}, \tilde{a}), \ldots \right\}$

44:             /* UMCTS update rule */

45:             $Q(\tilde{s}, \tilde{a}) \leftarrow \min\left\{ \dfrac{W(\tilde{s}, \tilde{a})}{N(\tilde{s}, \tilde{a})} + C \sqrt{\dfrac{\ln N(\tilde{s})}{N(\tilde{s}, \tilde{a})}}, U(\tilde{s}, \tilde{a}) \right\}$   $(*)$

46:         **end for**

47:     **end for**

48: **end for**

---

---

**Algorithm 3** UMCTS with Neural Network Environment Model

---

**Require:** MDPs $\{\mathcal{M}_1, \ldots, \mathcal{M}_M\}$, each with trained neural network parameters $\{\phi_1, \ldots, \phi_M\}$; A new MDP $M$ (partially known), with neural network $\Psi_\phi : \mathcal{S} \times \mathcal{A} \to \Delta(\mathcal{S})$; A distance function $\rho(\phi, \phi_i) \geq 0$ on parameter space (e.g., $\ell_2$-norm); Define $\hat{d}_{para}(M, M_i) = \rho(\phi, \phi_i)$; Lipschitz constants $L, L_2^{(i)}$, discount factor $\gamma$, $R_{\max}$, exploration constant $C$, iterations $T$; A default (simulation) policy $\pi$ for rollouts

---

1: **// For each task $M$ (with parameter $\phi$) run UMCTS**
2: Initialize root node $s_0$, counters $N(\cdot) = 0$, $N(\cdot, \cdot) = 0$, $W(\cdot, \cdot) = 0$
3: **for** $t = 1$ to $T$ **do**
4:     **Selection**:
5:       $s \leftarrow s_0$
6:     **while** all actions from s are expanded **and** s not terminal **do**
7:       $a \leftarrow \underset{a}{\operatorname{argmax}} \left( Q(s, a) \right)$
8:       $s \leftarrow$ child node after action $a$
9:     **end while**
10:     **Expansion**:
11:     **if** $s$ not terminal **then**
12:       choose an unvisited action $a_{\text{new}}$
13:       sample $s' \sim \Psi_\phi(\cdot \mid s, a_{\text{new}})$    // neural net predicts next state distribution
14:       create child node s'
15:       $N(s', \cdot) \leftarrow 0$, $W(s', \cdot) \leftarrow 0$
16:     **end if**
17:     **Simulation**:
18:       $G \leftarrow 0$
19:       $s_{\text{sim}} \leftarrow s'$
20:     **while** $s_{\text{sim}}$ not terminal **do**
21:       $a_{\text{sim}} \leftarrow$ sample from $\pi(\cdot \mid s_{\text{sim}})$
22:       // observe reward (possibly from real env or approximated by a learned reward model)
23:       $r_{\text{sim}} = R(s_{\text{sim}}, a_{\text{sim}})$
24:       $s_{\text{next}} \sim \Psi_\phi(\cdot \mid s_{\text{sim}}, a_{\text{sim}})$
25:       $G \leftarrow G + r_{\text{sim}}$
26:       /* update $\phi$ via gradient (e.g. supervised/unsupervised RL objective) */
27:       $\phi \leftarrow \phi - \eta \nabla_\phi \mathcal{L}\left( \phi; (s_{\text{sim}}, a_{\text{sim}}, s_{\text{next}}) \right)$
28:       $s_{\text{sim}} \leftarrow s_{\text{next}}$
29:     **end while**
30:     **Backpropagation**:
31:       traverse from $s'$ back to $s_0$
32:     **for all** visited state-action pairs $(\tilde{s}, \tilde{a})$ **do**
33:       $N(\tilde{s}) \leftarrow N(\tilde{s}) + 1$
34:       $N(\tilde{s}, \tilde{a}) \leftarrow N(\tilde{s}, \tilde{a}) + 1$
35:       $W(\tilde{s}, \tilde{a}) \leftarrow W(\tilde{s}, \tilde{a}) + G$
36:       // parametric distance to previously trained model $\phi_i$
37:       $\hat{d}_{para}(M, M_i) \triangleq \rho(\phi, \phi_i)$
38:       // Lipschitz-based upper bound
39:       $U_{\bar{\mathcal{M}}}(\tilde{s}, \tilde{a}) \leftarrow Q_{\bar{\mathcal{M}}}^*(\tilde{s}, \tilde{a}) + L \cdot \hat{d}_{para}(M, \bar{\mathcal{M}}) + L_2^{(i)}$
40:       $U(\tilde{s}, \tilde{a}) \leftarrow \min\left\{ \frac{R_{\max}}{1 - \gamma}, U_{\bar{\mathcal{M}}}(\tilde{s}, \tilde{a}), \ldots \right\}$
41:       // UMCTS update rule
42:       $Q(\tilde{s}, \tilde{a}) \leftarrow \min\left\{ \frac{W(\tilde{s}, \tilde{a})}{N(\tilde{s}, \tilde{a})} + C\sqrt{\frac{\ln N(\tilde{s})}{N(\tilde{s}, \tilde{a})}}, U(\tilde{s}, \tilde{a}) \right\}$    $(*)$
43:     **end for**
44: **end for**

---