# OpenReview forum: "Lipschitz-Guided Monte Carlo Tree Search with Knowledge Transfer across Sequential Tasks"
_ICLR.cc/2026/Conference — Submitted to ICLR 2026_

### Official Review · Reviewer_q4cU · 2025-10-24

**Soundness:** 2
**Presentation:** 3
**Contribution:** 2
**Rating:** 4
**Confidence:** 4

**Summary:**

The paper proposes LiZero, an MCTS variant for sequential task transfer. The key idea is an adaptive UCT (aUCT) bound that reuses past tasks’ Q-estimates with a Lipschitz-controlled penalty for task dissimilarity.

**Strengths:**

1. The problem is clearly formulated, and the scope is focused: an MCTS/UCT variant for lifelong task transfer.
2. Results across the task sequence show the effect cleanly with little gain on Task 1 and increasing gains by Task 10.
3. The high-level idea (planning-time transfer via Lipschitz control) is interesting and well-motivated.

**Weaknesses:**

1. Assumption in Theorem 4.3 (NN distance). The bound assumes a Lipschitz link between model outputs and a parameter-space metric. This is strong for modern networks; the constant L may be loose, and it is hard to verify. The usefulness of the method in higher-dimensional or continuous settings may depend on how valid this assumption is in practice.
2. Scalability. (1) Estimating d for many prior tasks can be expensive, especially with sampling-based estimators. This raises the need for an efficient and stable estimator. (2) Overall efficiency depends on the effectiveness of Theorem 4.3.
3. Given the above concerns, the paper would be more convincing if it reported results in an environment with a continuous state space (e.g., CartPole). It does not need to use continuous action spaces, since handling them is an orthogonal challenge for MCTS. Furthermore, it would help to demonstrate that the NN-based approach satisfies a Lipschitz relationship in a more complex environment with richer dynamics.

**Questions:**

1. Is the data-driven distance estimator the sampling-based approach?
2. Do MCTS baselines have access to true transition functions?
3. As described in the experiment setting, is the transition function non-deterministic? If so, how do you handle it during planning?
4. What is the number of MCTS iterations per decision in each experiment?

Minor issues / typos:
1. Can you fix the citep/citet in latex across the paper?
2. Duplicate citation of “Lipschitz lifelong reinforcement learning”
3. data-driven distance “estimater” -> estimator

---

> ### Author Response · Authors · 2025-11-20
>
> **Regarding the NN-based distance in Theorem 4.3 and its validity in higher dimensions**
>
> Regarding the Lipschitz assumption in Thm. 4.3, we note that explicitly Lipschitz-constrained neural networks are widely studied in robust learning and control. For example,Gouk et al. (Gouk et al., Regularisation of Neural Networks by Enforcing Lipschitz Continuity (Machine Learning, 2021) )  enforce global Lipschitz bounds via operator norms; Cissé et al.( Cissé et al., Parseval Networks: Improving Robustness to Adversarial Examples (ICML 2017).  ) constrain weight matrices so that the network is 1-Lipschitz; Anil et al.( Anil et al., Sorting Out Lipschitz Function Approximation (ICML 2019).   ) and Tsuzuku et al.( Tsuzuku et al., Lipschitz-Margin Training: Scalable Certification of Perturbation Invariance for Deep Neural Networks (NeurIPS 2018).   ) design and analyze architectures with certified Lipschitz constants, and spectral normalization has also been used to bound Lipschitz constants in deep RL ( Gogianu et al., Spectral Normalisation for Deep Reinforcement Learning (ICML 2021)  ). In such settings, the assumption of Thm. 4.3 is standard and verifiable, and LiZero-N can be directly applied; in general settings without such constraints, we recommend the assumption-light, sample-based LiZero-P estimator.
>
> **Regarding scalability with many prior tasks and dependence on Theorem 4.3**
>
> We have two methods for estimating the distance $d$, sample-based LiZero-P and NN-based LiZero-N.
>
> For LiZero-P, the distance estimator is exactly the sampling-based, data-driven approach defined in the paper: it reuses the transition samples generated by the search policy and applies an importance-sampling correction. Each new sample costs $O(1)$ extra operations to update a running estimate, so the complexity remains linear in the number of samples and is still dominated by the computation of the underlying MCTS.
>
> For LiZero-N, Theorem 4.3 is used to construct a parametric distance from model parameters. Here, the only dependence on the number of prior tasks $K$ is through computing $\hat{d}_{\text{para}}$
>
> for selected $k$’s, which costs
> $O( parameters)$ per comparison. In practice we do not need all $K$ tasks: aUCT only uses the minimum of the bounds, so it suffices to keep a small set of nearest neighbors in task space. Maintaining and updating such a set (e.g., the $K_0$ closest past tasks under $\hat{d}_{\text{para}}$ keeps the per-decision overhead bounded while preserving the guarantees.
>
> The complexity scales linearly with the number of samples and only mildly with the number of tasks if we use nearest-neighbor pruning. We will add a complexity discussion to clarify.

---

> > ### Author Response · Authors · 2025-11-20
> >
> > **Regarding continuous-state experiments and demonstrating the NN-based Lipschitz relation**
> >
> > We agree that adding at least one continuous-state environment  would make the empirical evidence more compelling, especially for the NN-based distance. Our theoretical framework already covers such settings: the MDP distance is defined for general (finite or continuous) state spaces, and Theorem 4.3 is stated directly for neural dynamics. The gridworld benchmark was chosen to (i) have exact access to tabular dynamics for computing the true distance and (ii) decouple the effect of aUCT from function-approximation error.
> >
> > |  | Task 1 |Task 2 | Task 3 |
> > |-----|-----|-----|-----|
> > | LiZero-N | 0.349 | 0.480 | 0.674 |
> > | Deep MCTS | 0.349 | 0.392 | 0.352 |
> >
> > We implemented our LiZero in a continuous state environment, MuJoCo Point Maze, and compared its performance to the baselines in our paper. The task is for a 2-degrees-of-freedom ball that is force-actuated in the cartesian directions x and y to reach a target goal in a closed maze. We generated a sequence of three tasks with randomly modified parameters and goal locations. Each task run for 2,000 epochs in MCTS. The following table shows our numerical results for these three tasks, while more complex experiments are still running and will be provided later. We see that our LiZero-N achieves increasingly better performance as more tasks are learned, i.e., from 22\% improvement over Deep MCTS baseline in the second task to 91\% improvement in the third task. This is demonstrates the ability if LIZero to achieve significant benefits in continuous state tasks — where MCTS process becomes more complex and thus aUCT bound via knowledge transfer achieves higher benefits. The results also show that LiZero-N can obtain reliable MDP distance estimate in these continuous state tasks, enabling aUCT for knowledge transfer.
> >
> > We will make this design choice explicit in the text and clearly state that evaluating LiZero-N on a continuous-state control task is a natural extension rather than a limitation of the theory itself. If space allows, we plan to add an additional continuous-state experiment in an extended version or the appendix; otherwise we will discuss this as future work and emphasize that the current results should be seen as a first, controlled validation.
> >
> > **Regarding the experimental setting, stochastic transitions, access to transitions, and MCTS budget**
> >
> > Data-driven distance estimator. Yes, the data-driven distance estimator is precisely the sampling-based estimator defined in the paper: it uses samples generated by the evolving search policy and an importance-sampling weight to approximate the MDP distance online.
> >
> > Transition function. The transition function we considered is indeed stochastic: each task samples a slip probability within a given range, so actions can lead to movement into unintended neighboring cells. During planning, both LiZero and the MCTS baselines simply sample the next state from this stochastic kernel at each simulated step, i.e., they perform standard Monte Carlo rollouts in a stochastic MDP.
> >
> > Access to the transition function for MCTS baselines. All MCTS algorithms (MCTS-R, MCTS-O, pUCT and LiZero variants) interact with the same generative environment: given a state–action pair, they obtain a successor state and reward by sampling from the environment’s transition kernel. No baseline uses additional structural information beyond this shared access; LiZero-U’s oracle distance uses the same tabular dynamics that define the environment and does not give LiZero extra information at planning time compared to the baselines.
> >
> >
> > Number of MCTS iterations per decision. In all experiments we use a fixed rollout budget $B$ MCTS simulations per decision, and the same $B$ is used for all compared methods to ensure fairness. We acknowledge that the exact value of $B$ is not currently stated in the paper; we will add this number explicitly in the experimental section and in the released configuration files so that the setup is fully reproducible.
> >
> > **Regarding minor issues**
> >
> > We will:
> >
> > Replace all citation commands by the appropriate \textbackslash citep/\textbackslash citet pattern and remove the duplicate citation of Lipschitz lifelong reinforcement learning.
> >
> > Fix the typo “estimater” $\rightarrow$ “estimator” and other minor typos.

---

### Official Review · Reviewer_ZSMC · 2025-10-31

**Soundness:** 2
**Presentation:** 3
**Contribution:** 3
**Rating:** 6
**Confidence:** 3

**Summary:**

This paper introduces LiZero that applies Monte Carlo Tree Search  to lifelong planning problems across a sequence of distinct MDPs. The core theoretical contribution is a new adaptive Upper Confidence Bound for Trees (aUCT), which enables knowledge transfer by balancing the similarity between tasks and the confidence of the stored knowledge. To make this practical, the paper presents efficient online algorithms to compute the aUCT bound using both data-driven and model-based approaches. Experimental results show that LiZero significantly outperforms existing MCTS and lifelong RL baselines, achieving faster convergence and greater sample efficiency.

**Strengths:**

- This paper has strength in theoretical development. Based on the definition of Lipschitz regularity across MDPs, the task distance d(M,M'),  and the uncertainty correction term P(N,N'), is directly instantiated in the design of the aUCT bound and the final action-selection rule. This mapping from theorem to implementation ensures that every part of the algorithm is mathematically grounded, enhancing the paper’s internal consistency and conceptual clarity

- In a sequence of ten gridworld tasks, LiZero consistently converges faster than both UCT-based MCTS variants (MCTS-R, MCTS-O, pUCT) and lifelong RL baselines (RMax, LRMax). The performance gain emerged in early tasks, indicating effective reuse of prior task knowledge. Quantitatively, LiZero achieves faster convergence speed and achieves early reward improvement. These results empirically validate the theoretical acceleration predicted by Theorem 3.4, confirming that tighter aUCT bounds facilitate faster and more stable planning across sequential tasks.

**Weaknesses:**

- While the authors demonstrate LiZero’s effectiveness by transferring knowledge across several related tasks, all experiments are performed within a single environment. This limited setup makes it difficult to assess the method’s generalizability. Moreover, despite repeatedly emphasizing the algorithm’s practicality (particularly in Section 4), the experiments are confined to a small 25 × 25 gridworld, which cannot be considered a practical or realistic scenario for lifelong planning with continuous states and/or actions. Detailed description of the experimental setup is missing.

- There is no sensitivity analysis for critical hyperparameters such as the exploration constant, the Lipschitz scaling coefficient, or the discount factor. The paper also does not analyze how search-level parameters that determine MCTS behavior (e.g., tree depth, branching width, rollout budget) affect performance. This absence of ablation and hyperparameter analysis prevents a clear understanding of which factors drive LiZero’s performance and whether the observed gains are robust across settings.

**Questions:**

(see the points of weakness above)

---

> ### Author Response · Authors · 2025-11-20
>
> **Regarding the experimental environment and generalizability**
>
> Our goal in this work is to provide a controlled evaluation where: (i) the full transition kernels and rewards of each task are available, so the MDP distance used in our theory can be computed exactly; and (ii) we can isolate the effect of the aUCT transfer term without confounding from function approximation. Note that, although all tasks live in the same abstract environment, each of the 10 tasks is a distinct MDP with modified transitions and/or rewards, so LiZero is already evaluated on a non-trivial sequence of different tasks.
>
> |  | Task 1 |Task 2 | Task 3 |
> |-----|-----|-----|-----|
> | LiZero-N | 0.349 | 0.480 | 0.674 |
> | Deep MCTS | 0.349 | 0.392 | 0.352 |
>
> We implemented our LiZero in a continuous state environment, MuJoCo Point Maze, and compared its performance to the baselines in our paper. The task is for a 2-degrees-of-freedom ball that is force-actuated in the cartesian directions x and y to reach a target goal in a closed maze. We generated a sequence of three tasks with randomly modified parameters and goal locations. Each task run for 2,000 epochs in MCTS. The following table shows our numerical results for these three tasks, while more complex experiments are still running and will be provided later. We see that our LiZero-N achieves increasingly better performance as more tasks are learned, i.e., from 22\% improvement over Deep MCTS baseline in the second task to 91\% improvement in the third task. This is demonstrates the ability if LIZero to achieve significant benefits in continuous state tasks — where MCTS process becomes more complex and thus aUCT bound via knowledge transfer achieves higher benefits. The results also show that LiZero-N can obtain reliable MDP distance estimate in these continuous state tasks, enabling aUCT for knowledge transfer.
>
> **Regarding sensitivity to hyperparameters and search-level parameters**
>
> We agree that a more systematic analysis of hyperparameters and search parameters would improve the empirical section. Conceptually, LiZero is designed so that the key parameters affect performance in a controlled way:
>
> The Lipschitz scaling coefficient (denoted for instance by $\lambda$) appears only as a multiplicative factor on the task-distance term inside aUCT. In the limit $\lambda \rightarrow 0$, this term vanishes and LiZero reduces exactly to the underlying UCT rule, so there is no risk of divergence or loss of correctness. Increasing $\lambda$ strengthens the influence of cross-task bounds; from the theory, the only way this can hurt is by making the bound too loose (large effective distance), in which case LiZero again behaves close to UCT. We will highlight this monotone and bounded effect in the text, which explains why performance is robust for a range of $\lambda$.
>
> The exploration constant in UCT enters in the standard confidence term. Our analysis of aUCT is compatible with any fixed positive value of this constant: the transferred term from past tasks and the within-task confidence term are simply added, and convergence of MCTS is preserved. We will clarify that, in all experiments, we use a single exploration constant (shared across LiZero and UCT baselines) and will report this value explicitly.
>
> The discount factor $\gamma$ is shared between LiZero and all baselines and is part of the MDP definition. The theoretical results (e.g., the value-difference bounds and the acceleration factor) hold for any $\gamma\in (0,1)$ and the dependence on $(1-\gamma)^{-1}$ is explicit in the constants; we will make this dependence visible in the main text to clarify how changing $\gamma$ scales the bound.
>
> For search-level parameters (tree depth, branching width, and rollout budget), LiZero uses exactly the same configuration as the UCT-based baselines; aUCT only modifies the selection rule at internal nodes.

---

### Official Review · Reviewer_WboF · 2025-11-01

**Soundness:** 3
**Presentation:** 2
**Contribution:** 4
**Rating:** 4
**Confidence:** 3

**Summary:**

This paper introduces LiZero, an extension of Monte Carlo tree search (MCTS) for transferring learned action-values from one task to another, motivated by lifelong planning. The paper provides theoretical results showing that the difference in the estimated action-values of two MDPs is upper-bounded by a distance between the MDPs and by a decreasing function of the visit counts. Based on this, they derive an upper bound for UCT, and they show that incorporating this upper bound into MCTS can speed up convergence. They also provide multiple practical methods for estimating the distance between MDPs in practice. Finally, they evaluate LiZero experimentally on a sequence of 10 tasks (benchmark from existing work), showing that LiZero's transfer provides a significant improvement over MCTS and lifelong RL baselines.

**Strengths:**

* LiZero is an interesting contribution, introducing the idea of transfer for lifelong learning in MCTS.
* The algorithm is supported by thorough theoretical analysis.
* The numerical results are positive.

**Weaknesses:**

While the paper introduces an interesting contribution supported by theoretical results, its presentation suffers from issues, some of which (e.g., inconsistent notation) make it difficult to verify the results of the paper:
* Proof of Theorem 3.2: Notation $n_1$ and $n_2$ seem to be undefined. Are these the same as $N$ and $N'$? In the same proof, $L_2$ also seems to be undefined. Is this the same as $P(N, N')$? In the same proof, the second line of Equation (18) seems to replace $\bar{G}$ with $G$ in the expectation (first term). Why is this an equality?
* Definition 3.1: The distance $d(\mathcal{M}, \mathcal{M}')$ also depends on $\kappa$. It would be helpful if this was clear from the notation.
* Theorem 3.2: Related to the previous point, the formulation of this theorem in the main text should include the $\kappa$ factor for the distance (which can only be inferred from the appendix).
* Theorem 3.4: On line 246, the definition of $\mathcal{S}_1$ includes $i$, but this symbol does not appear anywhere else in the formula, so it is not clear what the point of $\exists i$ is. Also, this definition is extremely different from the definition of $\mathcal{S}_1$ in Equation (20) in the appendix; not clear if these are equivalent. Further, this theorem in the main text uses notation $\mathcal{S}_0$, but this notation is undefined in the main text; it is only introduced in the proof. Without knowing what $\mathcal{S}_0$ is, the claim $\Gamma > 1$ on line 254 would not be clear.
* The paper does not seem to specify any hyperparameters used in the experiments (knowing these would be important for both the baselines and the proposed approach).

It would also be beneficial if the paper provided a few examples of practical applications.

A couple minor issues:
* Source code is available, but it has no comments or documentation, making it difficult for other to run it and reproduce the results.
* Section 5 should be broken up into subsection; or at the very least, add some paragraph headers, so that it is easier to find baselines, environment, etc.
* References should be formatted with \citep instead of \cite for readability. Quotation should be typeset as ``...''
* There are also a number of typos (e.g., line 077: "performance. faster", line 267: "MDPS") and editing mistakes (e.g., line 163: "(From Eqn 18)" seem misplaced).
 * The proposed approach is evaluated on a single series of tasks. It is not obvious how well the proposed approach will work on another set of tasks.

**Questions:**

Please clarify the notation (see weaknesses).

---

> ### Author Response · Authors · 2025-11-20
>
> **Proof of Theorem 3.2: $n_1,n_2,L_2$ , and Eq. (18)**
>
> Thank you for pointing out the notational inconsistencies. In the proof of Theorem 3.2:
> We will rename $n_1$ and $n_2$ to $N$ and $N^{\prime}$ respectively, to match the notation in the main text for the visit counts of the two tasks.
>
> The symbol $L_2$ in the appendix is intended to denote the Lipschitz factor arising from Definition 3.1; we will replace it by the same $\kappa$ (or $P(\mathcal{M},\mathcal{M}^{\prime})$, depending on the exact line) used in the theorem statement, to make the dependence on the MDP distance explicit and consistent.
>
> In Eq. (18), the replacement of $\bar{G}$ by $G$ inside the expectation is due to the fact that $\bar{G}$ is defined as the empirical return and $G$ as its underlying random variable under the trajectory distribution. We will unify the notation so that the same symbol is used in both places (i.e., we will not switch between $\bar{G}$ and $G$ in that step) and add a one-line comment that this step is simply taking the expectation of the empirical average.
>
> These changes make the proof fully consistent with the statement of Theorem 3.2 and avoid undefined symbols.
>
> **Regarding Definition 3.1, the role of $\kappa$, and Theorem 3.2:**
>
> $\kappa$ is a constant used to remove the mismatch in units between $\Delta R$ and $\Delta P$. We cannot add $\Delta R$ and $\Delta P$ directly, so we introduce $\kappa$ to make the sum well-defined. In Theorem 3.2 we analyze the Lipschitz property, and this analysis naturally determines the value of $\kappa$; it is not a tunable hyperparameter. The detailed derivation is given in Eq. (18).
>
> **Regarding experimental hyperparameters and reproducibility**
>
> In the revised version we will add a dedicated subsection summarizing all hyperparameters used for both LiZero and the baselines, including:
>
> MCTS parameters: exploration constant, rollout budget per decision, maximum depth, and stopping criteria.
>
> Lifelong RL baselines: learning rates, discount factor $\gamma$, target update frequency, network architectures, and replay buffer sizes.
>
> LiZero-specific parameters: choice of distance estimator (U/P/N), frequency of distance re-estimation, and any regularization or clipping used in practice.
>
> **Regarding practical applications and further evaluations on task sequences**
>
> We expect the results in this paper to be applicable to many practical problems that leverage MCTS for control and planning in time- or condition-varying environments, such as cyber security policy under different network conditions, control of power system with changing load characteristics, path planning in different terrains.
>
> |  | Task 1 |Task 2 | Task 3 |
> |-----|-----|-----|-----|
> | LiZero-N | 0.349 | 0.480 | 0.674 |
> | Deep MCTS | 0.349 | 0.392 | 0.352 |
>
> We implemented our LiZero in a continuous state environment, MuJoCo Point Maze, and compared its performance to the baselines in our paper. The task is for a 2-degrees-of-freedom ball that is force-actuated in the cartesian directions x and y to reach a target goal in a closed maze. We generated a sequence of three tasks with randomly modified parameters and goal locations. Each task run for 2,000 epochs in MCTS. The following table shows our numerical results for these three tasks, while more complex experiments are still running and will be provided later. We see that our LiZero-N achieves increasingly better performance as more tasks are learned, i.e., from 22\% improvement over Deep MCTS baseline in the second task to 91\% improvement in the third task. This is demonstrates the ability if LIZero to achieve significant benefits in continuous state tasks — where MCTS process becomes more complex and thus aUCT bound via knowledge transfer achieves higher benefits. The results also show that LiZero-N can obtain reliable MDP distance estimate in these continuous state tasks, enabling aUCT for knowledge transfer.
>
> **Regarding code documentation, Section 5 structure, references, and typos**
>
> We appreciate these detailed suggestions and will address them systematically:
>
> We will add comments and a short README to the released code, documenting how to run each experiment and how the options map to the LiZero variants and baselines.
>
> Section 5 will be split into subsections (e.g., “Environments”, “Baselines”, “Implementation Details”, “Results and Ablations”) so that readers can quickly locate specific information.
>
> We will switch to \textbackslash citep for all references that appear in parentheses and use proper LaTeX quotation marks ``…'' throughout.
>
> Finally, we will fix the typos you identified (“performance. faster”, “MDPS”, and the misplaced “(From Eqn 18)”) and run an additional pass to catch any remaining minor errors.

---

### Official Review · Reviewer_7szH · 2025-11-01

**Soundness:** 2
**Presentation:** 2
**Contribution:** 2
**Rating:** 4
**Confidence:** 3

**Summary:**

The paper proposes LiZero, a lifelong Monte Carlo Tree Search (MCTS) framework that transfers knowledge across sequential tasks via an adaptive UCT (aUCT) rule. The aUCT bound jointly depends on (i) the Lipschitz distance between tasks and (ii) the confidence of previously learned Q-values. Theoretical analysis shows improved sample efficiency with an acceleration factor > 1, and experiments on sequential grid-world tasks demonstrate 3–4 times faster convergence and ~31% higher early rewards compared to MCTS and lifelong RL baselines.

**Strengths:**

- Novel direction: First to formalize lifelong MCTS with cross-task transfer.

- Solid theory: Derives Lipschitz-continuity–based bounds with provable convergence and acceleration guarantees.

- Algorithmic contribution: Introduces both data-driven and neural distance estimators with theoretical error analysis.

- Empirical gains: Consistent, large improvements over MCTS, pUCT, and lifelong RL baselines.

**Weaknesses:**

- Restrictive assumptions: Relies on shared state/action spaces and Lipschitz-similar tasks, limiting generality beyond simple settings.

- Narrow evaluation: Only tested on discrete grid-worlds; scalability to continuous or high-dimensional domains remains unverified.

- Missing runtime analysis: The computational cost of distance estimation and maintaining multiple task models is not discussed.

- Weak Lipschitz enforcement: Theorem 4.3 assumes globally Lipschitz neural dynamics without explicit constraints (e.g., spectral norm control), which may render the bound loose in practice.

- Unvalidated distance metrics: The quality of the proposed distance estimators is never measured directly, leaving unclear whether performance gains truly arise from meaningful task similarity.

- Questionable IS practicality: The importance-sampling correction in Eq. (6) requires a known reference distribution U and full support of $\pi$; potential variance explosion and normalization issues are unaddressed.

- Unclear definition of U(s,a): Although a “normalized uniform distribution” is referenced, its concrete form and normalization method are unspecified, especially for large or continuous state spaces—raising reproducibility concerns.

**Questions:**

- How does LiZero behave when tasks are dissimilar or violate Lipschitz continuity—can it avoid negative transfer?

- What is the actual computational overhead of the distance estimation in LiZero-P/N?

- Could the framework integrate into MuZero-style model-based planning?

- How would performance scale with longer task sequences or partially overlapping state spaces?

- How are the importance weights $w_i$ obtained and stabilized in practice?

- How is the “normalized uniform distribution” U(s,a) implemented—over what domain, and are clipping or normalization strategies applied to control variance?

---

> ### Author Response · Authors · 2025-11-20
>
> **Regarding cases when tasks are dissimilar or violate Lipschitz continuity.**
>
> This is an interesting question. Since our work provides an upper-bound on $Q_{\mathcal{M}}^{N}(s,a)$ of the new task, it will become a loose upper-bound when the tasks are dissimilar or violate Lipschitz continuity. In the worst case, this loose upper-bound would not help improve the learning of the new task (using standard UCT), but will never result in a negative transfer. This can be seen at the top of Page 5, as we consider the minimum of the standard UCT bound and this $U_{\rm aUCT}(s,a)$ to solve the new task:
>
> $$a={\rm arg} \max_a \min \left[ \frac{W(s,a)}{N(s,a)} + C\sqrt{\frac{\ln N(s)}{N(s,a)}}, U_{\rm aUCT}(s,a)\right]$$
>
> Further details can be found in Theorem 3.4. The acceleration factor $\Gamma$ is defined as the ratio between the sample complexity of vanilla UCT and LiZero at a fixed accuracy level $\varepsilon$: $\Gamma = T_{\text{UCT}}(\varepsilon)/T_{\text{LiZero}(\varepsilon)}$. The proof shows that only state–action pairs in a set $\mathcal{S}_{1}$
>
>   where the transferred bound is strictly tighter than the standard UCT confidence term contribute to reducing $T_{\text{LiZero}}$. When tasks are dissimilar (large $d(\mathcal{M}_t,\mathcal{M}_k)$) or violate Lipschitz continuity, all such pairs drop out of $\mathcal{S}_1$, and the analysis yields $\Gamma = 1$, i.e., asymptotically no worse sample complexity than vanilla UCT. Thus LiZero in the worst case falls back to standard UCT in the absence of task similarity and Lipschitz continuity.
>
> **Regarding shared vs. partially overlapping state/action space.**
>
> All our results work with partially overlapping state/action space. For tasks with partially overlapping state/action spaces, we can consider the tasks with respect to a union of the individual state/action spaces, i.e., $\bar{\mathcal{S}}=\cup_i \mathcal{S}_i$ and $\bar{\mathcal{A}} = \cup_i \mathcal{A}_i$. Each task can be considered through its equivalent form on $\bar{\mathcal{S}},\bar{\mathcal{A}}$ by assigning zero transition probabilities and rewards to infeasible states/actions in $(\bar{\mathcal{S}}-{\mathcal{S}_i}, \bar{\mathcal{A}}-{\mathcal{A}_i})$. In this way, all our results in this paper remain unchanged for tasks with partially overlapping state/action spaces, including the distance $d(\mathcal{M},\mathcal{M}^{\prime})$, all bounds, and all algorithms. Of course, partially overlapping state/action spaces may imply smaller task similarity. We will move this discussion into the main text and make it explicit.
>
> **Regarding performance and scalability with longer task sequences.**
>
> Great question. Longer task sequences always help obtain an equal or better bound for knowledge transfer. Since each past task provides a new upper-bound $U_{\rm aUCT}(s,a)$, we can take the minimum of these upper-bounds to obtain the tightest bound for knowledge transfer. Thus longer task sequences help obtain an equal or better bound in our solution. Since computing $U_{\rm aUCT}(s,a)$ bounds from past tasks is much lower cost compared with MCTS algorithm itself, the potential computation cost of longer task sequences is insignificant compared to the benefits. This observation is indeed validated in our experiment results in Figure 1. Here we used all past tasks (in a sequence of 10 tasks) for knowledge transfer to a new tasks. Thus, Task 10 (which has a longer sequence) benefits from a tighter upper-bound than Task 2 (which has a shorter sequence), demonstrating superior improvement over the baselines in Figure 1. We will add this discussion.

---

> > ### Author Response · Authors · 2025-11-20
> >
> > **Regarding evaluation scope and scalability**
> >
> > We agree that the current experiments are restricted to discrete grid-worlds. This is deliberate: in this setting the full transition kernels and rewards are available, so we can (i) compute the exact MDP distance in Definition 3.1 and (ii) isolate the effect of the aUCT bound without approximation error from function approximation. We will emphasize this as a controlled first-step evaluation and explicitly acknowledge the limitation to discrete domains in the current experimental section.
> >
> > |  | Task 1 |Task 2 | Task 3 |
> > |-----|-----|-----|-----|
> > | LiZero-N | 0.349 | 0.480 | 0.674 |
> > | Deep MCTS | 0.349 | 0.392 | 0.352 |
> >
> > We implemented our LiZero in a continuous state environment, MuJoCo Point Maze, and compared its performance to the baselines in our paper. The task is for a 2-degrees-of-freedom ball that is force-actuated in the cartesian directions x and y to reach a target goal in a closed maze. We generated a sequence of three tasks with randomly modified parameters and goal locations. Each task run for 2,000 epochs in MCTS. The following table shows our numerical results for these three tasks, while more complex experiments are still running and will be provided later. We see that our LiZero-N achieves increasingly better performance as more tasks are learned, i.e., from 22\% improvement over Deep MCTS baseline in the second task to 91\% improvement in the third task. This is demonstrates the ability if LIZero to achieve significant benefits in continuous state tasks — where MCTS process becomes more complex and thus aUCT bound via knowledge transfer achieves higher benefits. The results also show that LiZero-N can obtain reliable MDP distance estimate in these continuous state tasks, enabling aUCT for knowledge transfer.
> >
> > At the same time, the distance $d(\mathcal{M},\mathcal{M}^{\prime})$ and the value-difference bounds in Theorem 3.2 are defined for general (finite or continuous) MDPs; the neural variant LiZero-N (Theorem 4.3) is specifically formulated for continuous/high-dimensional settings via parameter-space distances. Thus the theory is not tied to grid-worlds; only the empirical evaluation is.
> >
> > For MuZero-style planning, our framework is compatible with any MCTS variant that maintains task-specific action-value or value bounds. MuZero already learns a task-specific dynamics model $\Psi_{\phi_t}$. In a lifelong setting, one would keep models $\{\Psi_{\phi_k}\}_{k<t}$, compute distances $\hat{d}(\phi_t,\phi_k)$ (as in Theorem 4.3), and plug the resulting upper bounds into the pUCT selection rule in place of (or in addition to) the standard exploration bonus. This requires no new theory beyond replacing tabular $Q(s,a)$ by the MuZero value head; we will add a dedicated paragraph to clarify this extension and mark large-scale MuZero experiments as future work.
> >
> > **Regarding computational complexity and runtime overhead**
> >
> > We will add a short complexity / runtime paragraph:
> >
> > LiZero-P (data-driven, Eq. (6)/(7)) maintains a running estimator of $d$ using the same trajectories collected by MCTS; each new transition gives an $O(1)$ update to a few accumulators. Thus the runtime complexity depends on the number of trajectories to obtain distance estimate -- the sampling complexity is characterized in Theorems~4.1 and 4.2.
> >
> > LiZero-N (NN distance, Thm. 4.3) trains small dynamics networks per task on replay data already generated by search; computing $\hat{d}_{\text{para}}$ is just a norm in parameter space.
> >
> > Memory-wise, we only store Q-values for visited state-action pairs in past tasks and, optionally, compressed summaries or a subset of nearest tasks. We will include wall-clock comparisons (UCT vs LiZero-P/N) in the appendix to show that distance estimation adds only a small overhead relative to environment simulation and tree search.

---

> > > ### Author Response · Authors · 2025-11-20
> > >
> > > **Regarding Lipschitz enforcement in Theorem 4.3**
> > >
> > > Theorem 4.3 assumes that the learned dynamics model $\Psi_\phi$ is Lipschitz in its parameters with constant $L$, which yields an inequality of the form
> > >
> > > $$d(\mathcal{M},\mathcal{M}^{\prime}) \leq \hat{d}_{\text{para}} (\phi,\phi^{\prime}) = L ||\phi - \phi^{\prime}||$$
> > >
> > > This is used only to justify that plugging $\hat{d}_{\text{para}}$
> > >
> > >  into Theorem 3.2 produces a valid (possibly conservative) bound on cross-task value differences. Our convergence and acceleration guarantees for LiZero do not require that this bound be tight; if $\hat{d}_{\text{para}}$ overestimates the true distance, the resulting upper bounds in aUCT are looser, which can only reduce the effective transfer but cannot break correctness.
> > >
> > > In our experiments, we use standard networks without explicit spectral norm control, so we cannot claim a tight Lipschitz constant $L$ in practice. We will clarify that Theorem 4.3 should be read as a sufficient condition for any architecture with known Lipschitz constants (e.g., networks with spectral normalization or GroupSort activations). Incorporating such architectures to obtain numerically tighter bounds is orthogonal to the main contribution and left as future work.
> > >
> > > **Regarding distance estimators, their validation, and the IS scheme with $w_i$ and $U(s,a)$**
> > >
> > > Definition and implementation of $U(s,a)$. In all experiments, the state and action spaces are finite. We instantiate the normalized uniform distribution $U(s,a)$ as the uniform distribution on $\mathcal{S}\times \mathcal{A}$:$U(s,a) = 1/(|\mathcal{S}||\mathcal{A}|)$
> > > In Definition 3.1, $U$ is a distribution over triples $(s,a,s^{\prime})$; the practical estimator uses its $(s,a)$-marginal. We will state this explicitly in the main text to remove any ambiguity. For continuous domains, LiZero-P is not used in the current experiments; we will clarify that extending $U$ to such settings (e.g., via base densities and truncation) is left for future work.
> > >
> > > Importance weights $w_i$ and variance control. For LiZero-P, we set $w_i = U(s_i,a_i)/\pi(s_i,a_i)$, where $\pi$ is the empirical tree policy, estimated from visitation frequencies under the default policy. In the tabular grid-worlds, the UCT-style exploration ensures a strictly positive lower bound on $\pi(s,a)$ for every action at visited states, implying that $w_i$ are almost surely bounded. This prevents variance explosion in our setting and we did not observe numerical instability. For larger and/or continuous domains, we agree that more robust control is needed; standard remedies such as self-normalized importance sampling and weight clipping can be incorporated without changing the estimator’s unbiasedness or the theory. We will explicitly mention these options and adopt them in the released implementation.
> > >
> > > Validation of distance estimators. Our current ablations compare LiZero-U (oracle distance), LiZero-P (IS-based $\hat{d}_2$),
> > >
> > > and LiZero-N ($\hat{d}_{\text{para}}$) under the same aUCT machinery. The consistent ordering of performance
> > >
> > > $LiZero-U>LiZero-P>LiZero-N>MCTS-R$
> > >
> > > indicates that (i) using any distance estimator improves over no distance (MCTS-R), and (ii) tighter distances yield larger empirical acceleration, which is qualitatively aligned with Theorem 3.2. To provide a direct validation, we will add plots and statistics (e.g., correlation, calibration curves) comparing the true distance $d(\mathcal{M},\mathcal{M}^{\prime})$ and the estimated $\hat{d}_2$, $\hat{d}_{\text{para}}$ in the grid-worlds, thereby quantifying how accurately the estimators capture task similarity.

---

### Author Response · Authors · 2025-12-01

Dear AC,

Thank you for overseeing the review process of our paper under this unusual circumstance. Our work LiZero proposes a novel concept of adaptive UCT (aUCT) to transfer knowledge in Lipschitz lifelong MCTS context, for solving a sequence of varying MDPs. This has not been studied in prior work. Our paper also analyzes LiZero's acceleration factor in terms of improved sampling efficiency and develops efficient algorithms to compute aUCT in an online fashion. LiZero outperforms baselines by 3-4x speedup in convergence, when solving a sequence of MDPs.

During the rebuttal phase, we have conducted additional experiments to address questions raised by the reviewers (including executing LiZero on a sequence of continuous state tasks) and clarified several misunderstandings. We wanted to provide a brief summary here for your reference. We note that all four reviewers found LiZero to be a novel and interesting contribution on lifelong MCTS with cross-task transfer. In particular, Reviewer WboF rated the contribution as excellent and emphasizes that the algorithm is “supported by thorough theoretical analysis,” while Reviewer ZSMC highlighted the clean mapping from Lipschitz-regularity theory to the aUCT selection rule. ALl reviewer stated they would be comfortable with acceptance.

**Robustness of transfer.** Potentially negative transfers mentioned by Reviewer 7szH is an interesting point, but is guaranteed not to occur in our LiZero solution. This is because that our our aUCT rule always picks $\max_{a}\min\{UCT\mathrm{term},U_{aUCT}(s,a)\}$so when tasks are dissimilar, the transferred bound becomes loose and LiZero provably falls back to standard UCT. This is also reflected in our analysis in Theorem 3.4, which yields an acceleration factor $\Gamma\geq 1$, with equality in the worst case—as LiZero falls back to vanilla UCT.

**Tasks with partially overleaping state/action spaces.** As we explained in the paper, our LiZero readily works in this case by embedding all tasks into a union state/action space, without having to change any results, bounds, or algorithms in the paper. Some of our evaluation tasks assign a zero transition probability to certain state-action pairs, which is equivalent to partially overleaping state/action spaces and demonstrates the effectiveness of our algorithm in this case. We have provided further clarifications in the reponse.


**Experimental evidence including continuous-state tasks**

In the original evaluation results using 10-task sequences, all reviewers recognized that LiZero shows consistent and substantial speed-up over UCT-based MCTS (MCTS-R/O, pUCT) and lifelong RL baselines (RMax, LRMax), especially in later tasks where more prior knowledge is available—exactly matching the predicted acceleration behavior in Theorem 3.4.

To address questions regarding continuous-state tasks, we added a new experiment using a sequence of continuous tasks from MuJoCo Point Maze (involving a continuous-force actuated mass). LiZero-N matches Deep MCTS on Task 1 (0.349 vs 0.349) and achieves $\simeq$22\% improvement on Task 2 (0.480 vs 0.392) and $\simeq$91\% improvement on Task 3 (0.674 vs 0.352). This demonstrates that the proposed distance-guided transfer remains effective in continuous-state domains where MCTS is more expensive, strengthening the practical impact of our results. we conducted new experiments in main manuscript Figure 3.

**Main contribution and theory.**
LiZero is, to the best of our knowledge, the first framework to formalize lifelong MCTS with a new adaptive UCT (aUCT) bound that transfers value estimates across tasks under a Lipschitz-controlled MDP distance. We made several clarifications in the detailed responses to reviewers' questions. Theoretical results (Def. 3.1, Thm. 3.2, Thm. 3.4) give explicit value-difference bounds and a provable acceleration factor. Reviewer WboF explicitly notes that the algorithm is backed by “thorough theoretical analysis,” which directly addresses this question.

**Clarifications and presentation fixes**
We will fix all noted notation issues (e.g.$n_1,n_2,L_2,\mathcal{S}_0,\mathcal{S}_1$) , unify them with the main text, and add a dedicated subsection listing all hyperparameters and MCTS budgets for full reproducibility. We also commit to improving Section 5 structure, citation style, and code documentation as requested.

Overall, our work makes a novel and technically rigorous contribution, with new experiment results addressing questions on continuous state/action spaces. We respectfully appreciate the AC’s consideration during the meta-review and hope our work’s impact and value to the ICLR community will be recognized. We have also uploaded an updated version of the main manuscript, where all newly added content is highlighted in blue for ease of reference.

---

### Meta-Review · Area_Chair_Bhne · 2026-01-13

**Summary:**

Reviewers agreed that LiZero is an interesting attempt to formalize lifelong MCTS via an adaptive UCT bound that transfers knowledge across a sequence of MDPs, with a generally solid theoretical story and promising empirical gains (ZSMC; WboF; 7szH; q4cU). The main concerns shaping the decision were (i) limited empirical scope and unclear scalability beyond a single discrete gridworld benchmark (7szH; ZSMC; q4cU), (ii) the strength and practical verifiability of the Lipschitz assumptions used for the neural-distance variant (q4cU; 7szH), (iii) missing or under-specified implementation details affecting reproducibility (e.g., MCTS budget, hyperparameters, and the definition/implementation of the IS reference distribution) (7szH; WboF; ZSMC), and (iv) presentation and notation inconsistencies that made the theory harder to check (WboF).

**Reviewer Concerns:**

Most concerns were partially addressed through clarifications and limited new evidence. The risk of negative transfer was largely resolved by the argument that aUCT safely falls back to standard UCT when tasks are dissimilar, and partially overlapping state/action spaces were clarified via a union-space formulation (7szH). Limited generalization beyond gridworlds remains a concern, although the added continuous-state PointMaze results and the framing of experiments as controlled evaluations mitigate this issue somewhat (7szH; ZSMC; q4cU). The strong Lipschitz assumptions for the neural-distance variant were acknowledged as sufficient but potentially loose, leaving their practical validity in deep models as future work (q4cU; 7szH).

**Reviewer Scores:**

(7szH) would likely move from 4 to 5, due to the no-negative-transfer guarantee and added continuous-state evidence.

(WboF) would likely move from 4 to 5, assuming the promised fixes to notation and reproducibility.

(ZSMC) would likely remain around 6, as robustness and sensitivity analyses remain limited.

(q4cU) would likely move from 4 to 5, with scaling clarified but NN Lipschitz concerns persisting.

---

### Decision · Program_Chairs · 2026-01-26

Reject